# A novel *SLC2A10* gain-of-function variant links glycolytic macrophage polarization to chronic nonbacterial osteomyelitis

Xiang Li, Lihang Shen, KuanKuan Jia, Shuo Chen, Jingang An

Chronic nonbacterial osteomyelitis is a rare autoinflammatory bone disorder with unclear etiology. In a Chinese family, whole-exome sequencing identified a heterozygous missense variant in *SLC2A10* that cosegregated with disease. Structural modeling suggested that the substitution disrupts hydrogen bond within the transmembrane domain, potentially altering glucose transport activity. In vitro, the expression of the p.Asp292Glu variant in macrophages enhanced glucose uptake, increased glycolytic activity, and promoted pro-inflammatory polarization, accompanied by increased production of inflammatory mediators. The variant also accelerated osteoclast differentiation and suppressed osteoblast-mediated mineralization, indicating a disruption of bone homeostasis. Zebrafish expressing the p.Asp292Glu variant exhibited enhanced inflammatory responses, increased reactive oxygen species accumulation, elevated macrophage infiltration in the mandibular region, and impaired bone mineralization. Transcriptomic analysis confirmed activation of glycolytic and inflammatory signaling pathways, as well as enrichment of osteoclast differentiation networks. These findings demonstrate that the *SLC2A10* variant exerts a gain-of-function effect driving glycolysis-dependent macrophage polarization and osteoclastogenesis, leading to chronic inflammatory bone destruction. This study broadens the genetic understanding of chronic nonbacterial osteomyelitis and suggests a potential therapeutic target.

## Introduction

Chronic nonbacterial osteomyelitis (CNO) is an autoinflammatory condition of the bone that mainly occurs in children and teenagers, presenting with bone pain, localized swelling, and, in some cases, local warmth over affected bones (Hofmann et al, 2017). Associated dermatological manifestations include palmoplantar pustulosis, psoriasis, and acne, whereas noninfectious arthritis occurs in up to 30% of patients (Borzutzky et al, 2012; Schnabel et al, 2016). The clinical spectrum ranges from single bone lesions to recurrent, multifocal inflammation involving multiple bones, termed chronic recurrent multifocal osteomyelitis (Ferguson & Sandu, 2012). Although uncommon, mandibular involvement has been reported in patients with CNO, typically presenting with repeated episodes of swelling, pain, and trismus, but without abscess development or fistula formation, and radiological features may include mixed sclerosis and lysis, subperiosteal bone formation, and cortical discontinuity (Hofmann et al, 2017; Jia et al, 2023).

Autoinflammatory bone diseases represent a distinct category of disorders caused by unprovoked activation of the innate immune system, leading to sterile bone inflammation (Stern & Ferguson, 2013). These lesions are culture-negative and lack identifiable pathogens on histopathology (Girschick et al, 2007; Catalano-Pons et al, 2008; Borzutzky et al, 2012). Increasing evidence indicates that genetic factors are critical drivers of these diseases. This is supported by the identification of variants in *LPIN2* (Majeed syndrome) and *IL1RN* (deficiency of the interleukin-1 receptor antagonist) that cause autoinflammatory bone lesions (Ferguson et al, 2005; Aksentijevich et al, 2009; Reddy et al, 2009), as well as analogous phenotypes observed in animal models with similar genetic defects (Ferguson et al, 2006; Grosse et al, 2006).

Here, we investigated a Chinese family affected by autosomal dominant CNO of the mandible. In this family, we detected a rare heterozygous missense change within *SLC2A10* (NM_030777.4), c.876C>A (p.Asp292Glu), a gene encoding the glucose transporter 10 (GLUT10), in which pathogenic variants have been reported to cause arterial tortuosity syndrome (Seidner et al, 1998; Coucke et al, 2006). Although the vascular role of GLUT10 is established, its function in bone metabolism remains unexplored. The aim of this study was to elucidate the pathogenic role and underlying mechanism of the *SLC2A10* variant in CNO of the mandible. Through integrated clinical, genetic, and functional analyses, including in vitro macrophage and osteoclast assays and in vivo zebrafish modeling, we demonstrate that this variant induces glycolysis-dependent macrophage polarization and osteoclastogenesis, thereby linking metabolic reprogramming to chronic inflammatory

Department of Oral and Maxillofacial Surgery, Peking University School and Hospital of Stomatology, Beijing, China

Correspondence: jkjiakuan@163.com; chens@bjmu.edu.cn; anjingang@126.com

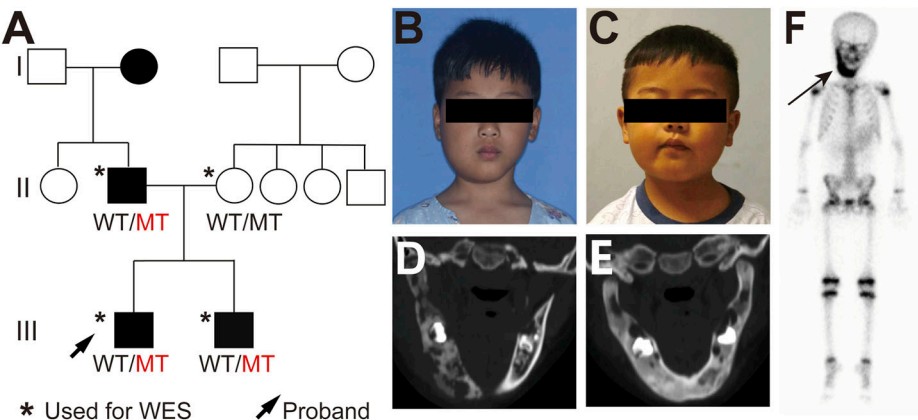

**Figure 1. Oral clinical characteristics and analysis of the solute carrier family 2 member 10 (*SLC2A10*) variant in individuals with chronic nonbacterial osteomyelitis (CNO).**
**(A)** Pedigree of the family with CNO. The black arrow indicates the proband. Squares and circles represent males and females, respectively; filled symbols indicate affected individuals. Asterisks denote individuals who underwent whole-exome sequencing. **(B)** Patient III-1's facial photo shows an 8-yr-old boy with swelling and pain on the right side of his mandible and limited mouth opening. **(C)** Patient III-2's facial photo shows a 4-yr-old boy with swelling and pain on the right side of his mandible and limited mouth opening. **(D)** Computed tomography (CT) image of patient III-1 shows bone destruction in the right mandible, with discontinuous cortical bone. **(E)** CT image of patient III-2 shows a mixture of mandibular bone destruction and sclerosis. **(F)** Whole-body bone scan of patient III-1 with CNO; the black arrow indicates the accumulation of nuclei in the mandibular region, indicating active bone metabolism and inflammation in this area.

bone destruction. We provide novel evidence that enhances the current understanding of bone metabolism in CNO.

# Results

## Clinical characteristics and variants identified in CNO families

We analyzed an independent pedigree with CNO. In this family 1, whole-exome sequencing identified a heterozygous missense variant in *SLC2A10*, c.876C>A (p. Asp292Glu) (Figs 1A and S1). This variant was classified according to ACMG guidelines (likely pathogenic).

Clinically, two affected children (III-1 and III-2) presented with recurrent facial pain, swelling, and limited mouth opening (Fig 1B and C). Imaging studies revealed osteolytic lesions and irregular cortical bone in the mandible (Fig 1D and E). Bone scintigraphy of patient III-1 demonstrated abnormal radionuclide accumulation in the mandibular region, consistent with typical CNO manifestations (Fig 1F). The proband's father (II-1) and grandmother (I-1) also exhibited skeletal tenderness and severe cutaneous acne.

Importantly, the *SLC2A10* variant segregated with the disease phenotype among the tested members of this family, supporting its potential involvement in CNO and suggesting a possible association with inflammatory bone manifestations.

## Identified variant and pathogenic analysis

Sanger sequencing verified the presence of the heterozygous variant c.876C>A in all affected family members (III-1, III-2, II-1), whereas it was absent in unaffected individuals (II-2) (Fig 2A). The variant, located in exon 2 of *SLC2A10* (chr20q13.12), substitutes aspartic acid with glutamic acid at position 292. Cross-species sequence alignment revealed that Asp292 is highly conserved among mammals, indicating functional importance (Fig 2B). Structural modeling suggested that in the WT GLUT10 protein, D292 is predicted to form hydrogen bonds with adjacent transmembrane residues. The D292E substitution is predicted to alter two hydrogen bond interactions, which may influence local structural stability (Fig 2C).

## Association between inflammatory pathway activation and abnormal osteoclast proliferation

Histological examination of mandibular bone tissue from CNO patients using HE staining (Fig 3A) and tartrate-resistant acid phosphatase (TRAP) staining (Fig 3B), revealed a marked increase in TRAP-positive osteoclasts compared with healthy controls, accompanied by disorganized trabecular structure and abundant resorption pits. HE staining further showed localized inflammatory cell infiltration and bone matrix degradation, consistent with an overactivated osteoclast phenotype.

Protein microarray profiling of mandibular bone tissues from CNO patients demonstrated markedly elevated levels of inflammatory mediators, including IL-6, TNF-α, and IL-1β, compared with controls (Fig 3C). Heatmap visualization revealed a distinct cytokine expression pattern in CNO lesions, consistent with the presence of a pro-inflammatory microenvironment. In summary, these findings demonstrate that CNO lesions are associated with heightened inflammatory activity and increased osteoclast numbers at the affected sites.

## *SLC2A10* variant enhances inflammatory activity in macrophages

Within the innate immune system, macrophages are crucial mediators of inflammation. Macrophage activation is generally polarized into an M1 phenotype that drives inflammatory responses or an M2 phenotype that mediates anti-inflammatory functions. Maintaining equilibrium between these states is vital for adequate immune defense without triggering excessive inflammation. In autoinflammatory diseases, this equilibrium is disrupted, leading to sustained inflammatory activation and disease progression.

To investigate the effect of *SLC2A10* variant on macrophages, RAW264.7 cells were transduced with lentiviral vectors encoding WT *SLC2A10* or the c.876C>A (p.D292E) variant. Western blot analysis

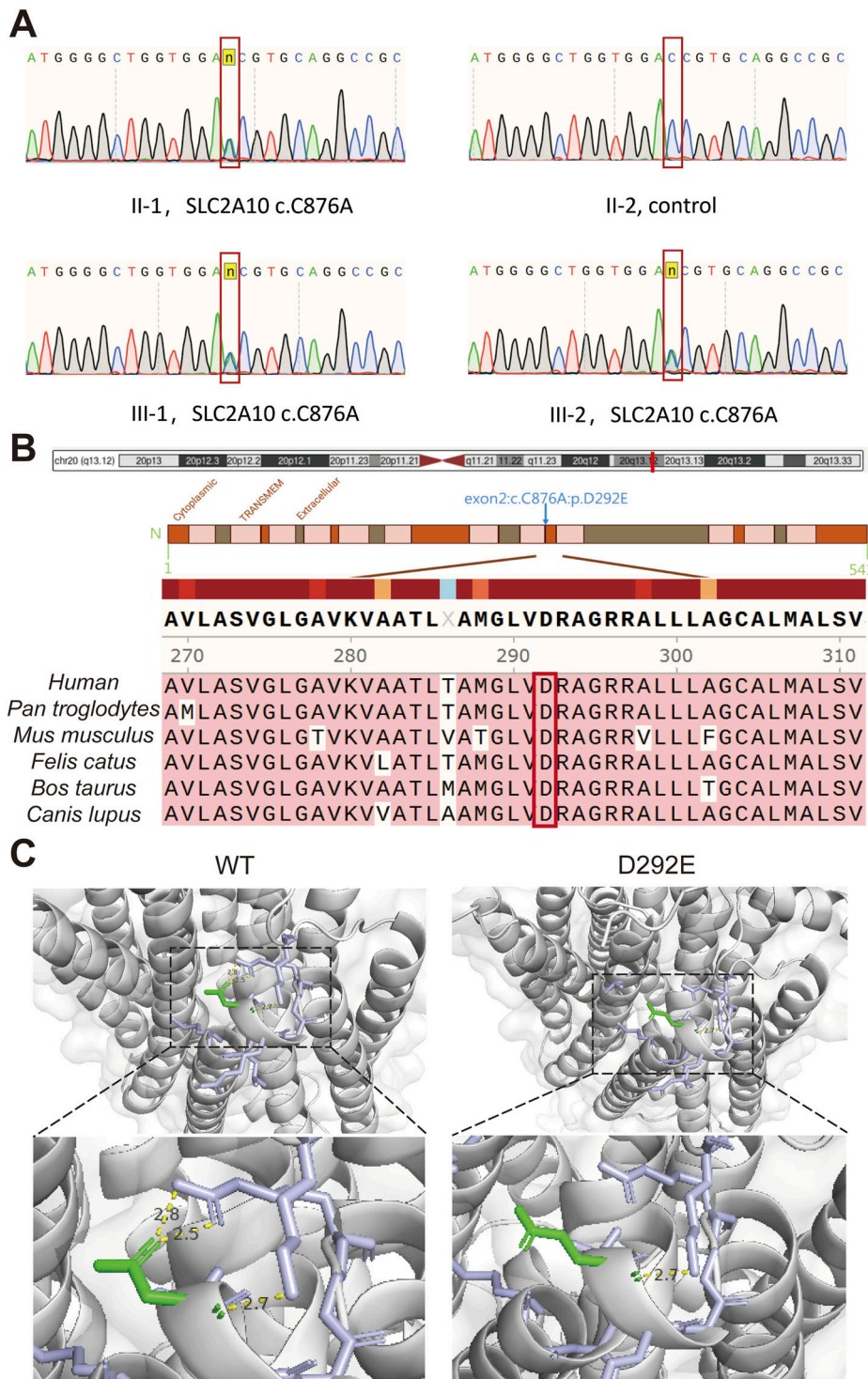

**Figure 2. Mutation of *SLC2A10* in individuals with CNO.**
**(A)** Sanger sequencing results showing the c.876C>A mutation in *SLC2A10* in Family 1. Red boxes indicate the mutated site in patients compared with controls. **(B)** Genomic location and amino acid change of the *SLC2A10* variant. Multispecies alignment of *SLC2A10* protein sequences was performed using SnapGene 6.0.2. The degree of conservatism of amino acids is shown as a color block (red: conserved; blue: not conserved, *SLC2A10* missense mutations at a highly conserved site). **(C)** Three-dimensional structural model of the *SLC2A10* protein. In the WT GLUT10 protein, residue D292 forms stable hydrogen bonds with adjacent transmembrane residues. The D292E substitution disrupts two of these critical hydrogen bonds. (The position of amino acid residue 292 is shown in green.) WT, wild type; D292E: aspartic acid-to-glutamic acid substitution at position 292 in GLUT10.

revealed that GLUT10 protein levels were slightly higher in macrophages overexpressing D292E than in WT cells (Fig S2A). In addition, glucose uptake assays showed that after knockout of *SLC2A10* in macrophages, glucose uptake rate was significantly higher in D292E-rescued cells than in WT cells, suggesting that this may be a gain-of-function mutation (Fig S2B). RT–qPCR analysis

found that LPS-stimulated macrophages carrying the variant exhibited significantly elevated mRNA expression of pro-inflammatory factors *TNF-α*, *IL-1β*, *IL-6*, and *iNOS* compared with WT controls (Fig 4A). Western blot revealed a marked up-regulation of iNOS protein levels in the D292E group after LPS induction, indicating that the variant promotes macrophage M1 polarization (Fig 4B). ELISA

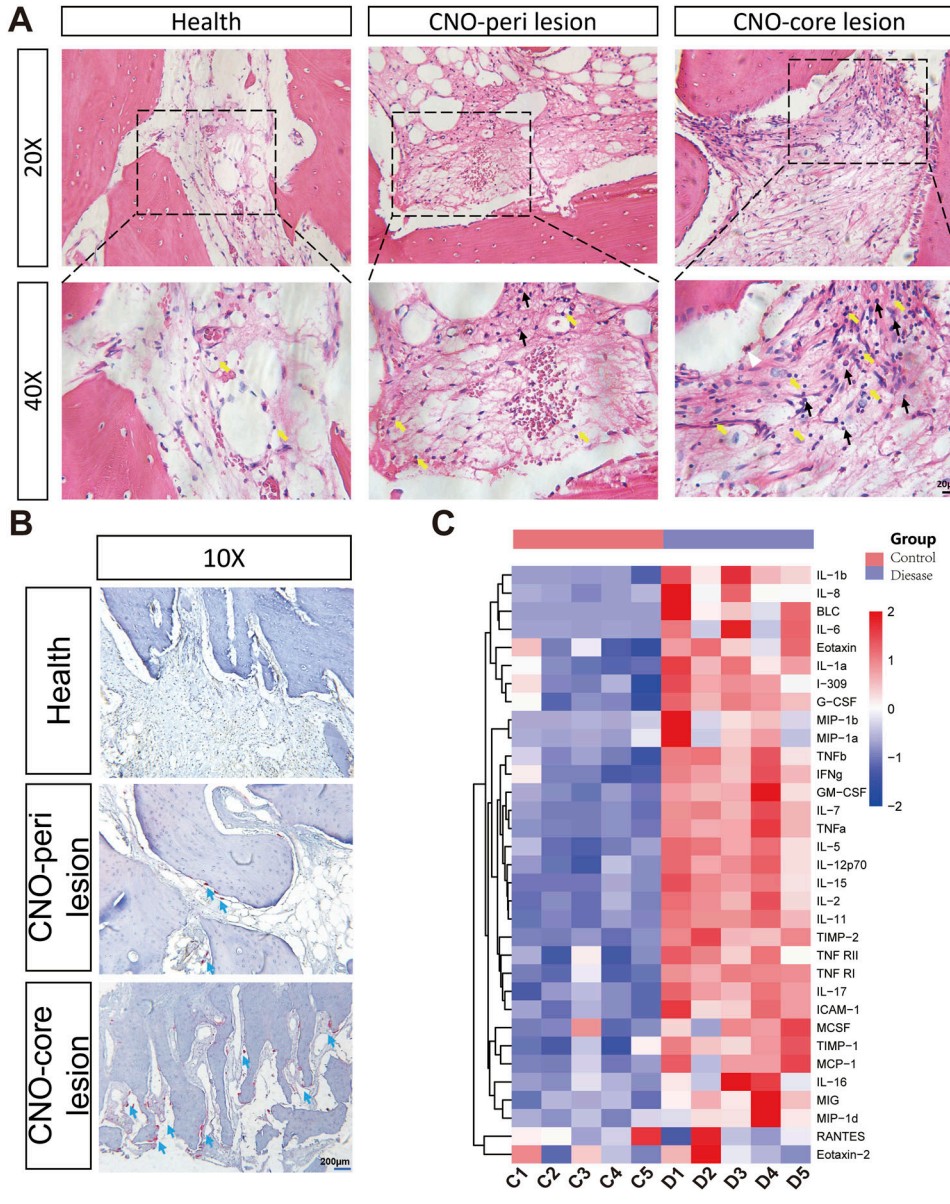

**Figure 3. Increased inflammatory activity and osteoclast accumulation in CNO lesions.**
**(A)** Histopathological evaluation by hematoxylin and eosin (H&E) staining demonstrates progressive inflammatory infiltration and osteoclast accumulation from (1) normal mandibular tissue of healthy controls to (2) the perilesional region and (3) the core lesion area. White arrows indicate multinucleated giant cells; black arrows indicate plasma cells; yellow arrows indicate lymphocytes. **(B)** TRAP staining identifies osteoclasts. Blue arrows indicate osteoclasts. Quantitative comparison among normal mandible, perilesional tissue, and lesional mandible revealed a significantly increased number of osteoclasts in the core lesion area. **(C)** Heatmap showing the expression profiles of 33 inflammatory mediators in mandibular tissues from five CNO patients and five healthy controls. Colors represent relative expression levels (red, high; blue, low).

further confirmed that at the protein level, IL-1$\beta$, IL-6, and TNF-$\alpha$ were significantly increased in D292E cells (Fig 4C). These findings demonstrate that the variant enhances inflammatory responses at both transcriptional and translational levels.

Flow cytometry analysis showed that LPS stimulation significantly increased reactive oxygen species (ROS) production in D292E cells (Fig 4E and F), whereas the proportion of CD86-positive cells, a marker of M1 polarization, was also elevated (Fig 4D), providing additional evidence for the variant's pro-inflammatory effect.

To explore the underlying mechanism, we examined the activation status of the NF-$\kappa$B signaling pathway. Western blot revealed that LPS stimulation markedly increased phosphorylation of p65 (an NF-$\kappa$B subunit) and I$\kappa$B$\alpha$ in D292E cells (Fig 4G), suggesting that the variant may exert its pro-inflammatory effects through activation of NF-$\kappa$B signaling.

To minimize endogenous effects, we further validated our findings in *SLC2A10*-knockout macrophages (Fig S3A). We found that the variant promotes increased secretion of inflammatory cytokines and enhances M1 polarization in macrophages, likely through activation of the NF-$\kappa$B signaling pathway (Figs S2D and S3B, C, and E).

### *SLC2A10* variant promotes osteoclast differentiation and inhibits osteogenesis

To explore the effect of variants on macrophage–osteoclast differentiation, we also overexpressed WT and D292E *SLC2A10* in RAW264.7 cell lines. RT–qPCR analysis revealed that after 3 d of RANKL induction in RAW264.7 cells, the D292E group exhibited significantly higher mRNA expression of osteoclast differentiation markers,

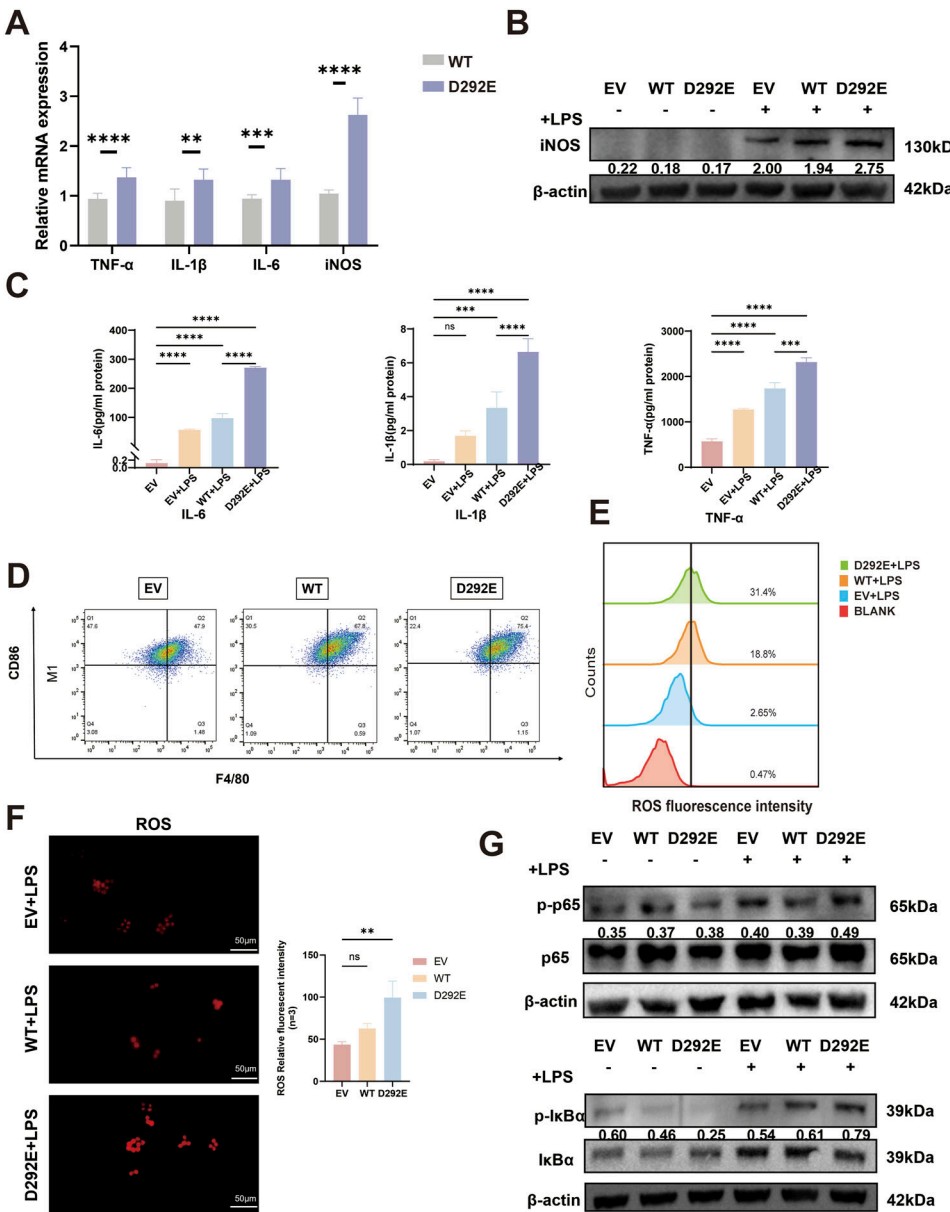

**Figure 4. *SLC2A10* variant drives M1 macrophage polarization in RAW264.7.**
**(A)** mRNA expression of inflammatory factors at 12 h after LPS stimulation in WT and D292E. **(B)** Western blot analysis of iNOS expression in RAW264.7 cells induced by LPS. Relative gray value analysis of Western blot. **(C)** ELISA results showing the secretion levels of inflammatory factors in the supernatant of RAW264.7 cells after LPS induction for 24 h. (EV, empty vector cells; WT, overexpression of WT *SLC2A10* cells; D292E, overexpression of variant *SLC2A10* cells) **(D)** Flow cytometry results at 24 h after LPS stimulation. **(E)** Flow cytometry analysis of reactive oxygen species at 24 h after LPS stimulation. **(F)** Immunofluorescence detection of reactive oxygen species at 24 h after LPS stimulation. **(G)** Western blot showing phosphorylated marker proteins for NF-κB pathway activation of p65 and IκBα after LPS stimulation. N = 3 per group. One-way ANOVA was used to evaluate the significance between multiple groups, and a *t* test was used to calculate the *P*-value for post hoc comparisons. All data are shown as the mean ± SD; *$P < 0.05$, **$P < 0.01$, ***$P < 0.001$, ****$P < 0.0001$.

including *CTSK* and *TRAP*, compared with the WT group (Fig 5A). Western blot further demonstrated that protein expression of CTSK and TRAP was markedly increased in the D292E group relative to the WT group after 6 d of RANKL induction (Fig 5B). TRAP staining after 6 d of RANKL induction revealed the formation of characteristic vacuolated multinucleated osteoclast giant cells in the D292E group. Quantitative analysis confirmed a statistically significant increase in osteoclast numbers compared with the WT group (Fig 5C and D). These results indicate that D292E cells possess enhanced osteoclast differentiation capacity. To exclude potential confounding effects from endogenous *SLC2A10* expression, we further performed rescue experiments in *SLC2A10*-deficient cells. Reintroduction of the D292E variant led to increased expression of osteoclast effector molecules compared with controls (Fig S3D). To

further explore the underlying mechanism, we next examined signaling pathways involved in osteoclast differentiation. The mutant appeared to promote osteoclast differentiation through activation of the ERK-AP-1 signaling axis (Fig S3F and G).

To assess osteoclast–osteoblast interactions, MC3T3 cells (osteogenically induced for 1 wk) were cocultured with RAW264.7 cells (RANKL-induced) from the Vector, WT, and D292E groups. RT–qPCR analysis showed that coculture significantly suppressed osteogenesis-related gene expression, with the D292E group exhibiting a more pronounced inhibitory effect than the WT group (Fig 5E). Western blot further confirmed that after 2 wk of coculture, RAW264.7 cells expressing the c.876C>A variant significantly suppressed osteogenic marker expression in MC3T3 cells compared with the other experimental groups (Fig 5F and I).

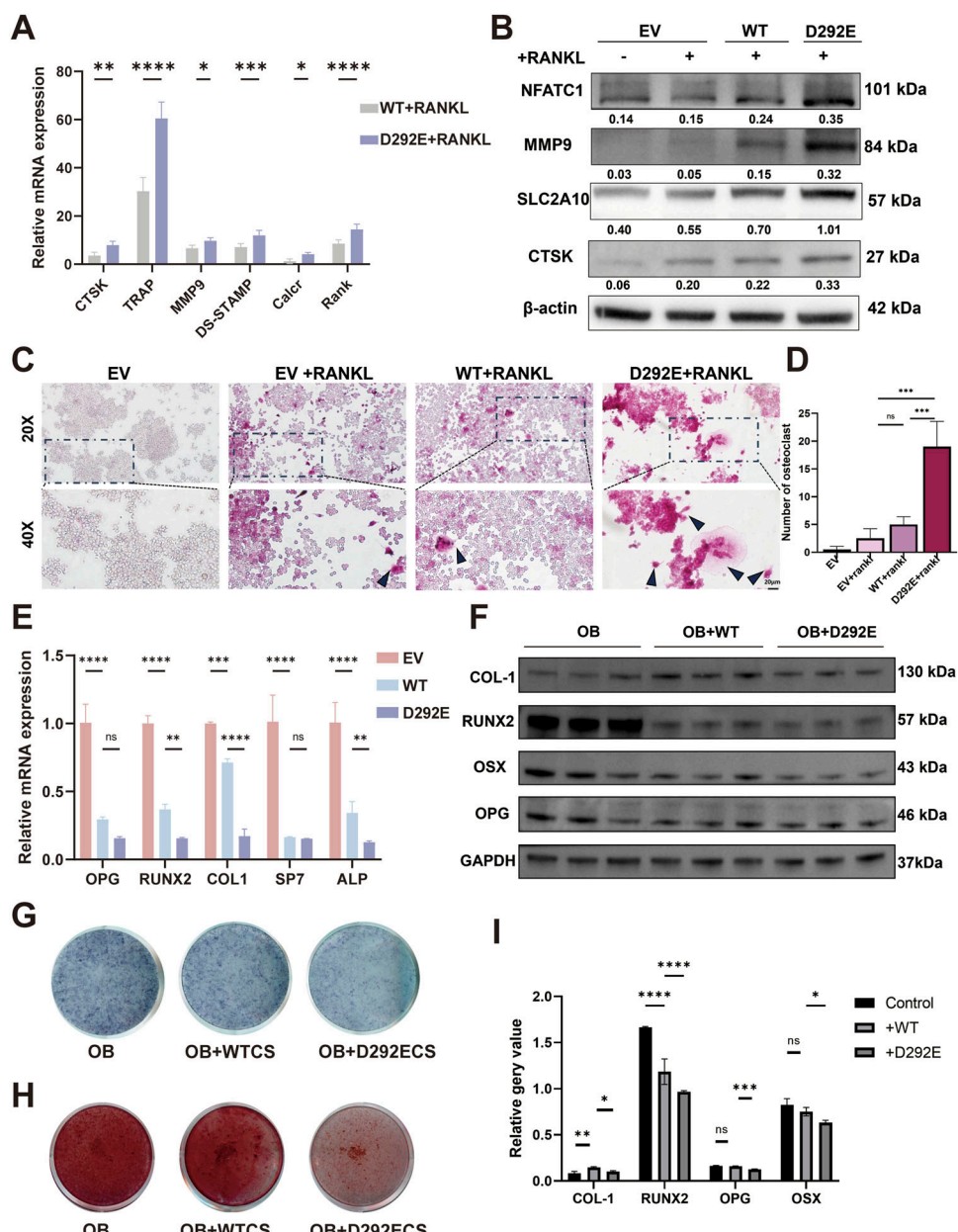

**Figure 5.** *SLC2A10* mutation promotes macrophage differentiation into osteoclasts.

**(A)** mRNA expression of osteoclast factors at 3 d after RANKL induction. **(B)** Western blot analysis of osteoclast protein expression at 5 d after RANKL induction. **(C)** TRAP staining marks multiple groups of osteoclasts at 1 w after RANKL induction; arrows indicate fused multinucleated giant cells. **(C, D)** Quantification of osteoclast numbers in (C). **(E)** mRNA expression of osteoblast factors at 3 d after coculture of RAW264.7 induced by RANKL and MC3T3-E1 induced by osteogenesis. **(F)** Western blot analysis of osteoblast protein expression after 5-d coculture of RAW264.7 induced by RANKL and MC3T3-E1 induced by osteogenesis. **(G)** Detection of mineralization effect by alkaline at 1w after coculture of RAW264.7 induced by RANKL and MC3T3-E1 induced by osteogenesis. **(H)** Detection of mineralization effect by Alizarin Red staining at 2w after coculture of RAW264.7 induced by RANKL and MC3T3-E1 induced by osteogenesis. **(F, I)** Quantification of results in (F). NC, RAW264.7; WT, wild type; MUT, mutant; CS, cell supernatant. N = 3 per group. One-way ANOVA was used to evaluate the significance between multiple groups, and a *t* test was used to calculate the *P*-value for post hoc comparisons. All data are shown as the mean ± SD; *$P < 0.05$, **$P < 0.01$, ***$P < 0.001$, ****$P < 0.0001$.

ALP and Alizarin Red staining demonstrated that MC3T3 cells cocultured with the D292E group displayed further reduced osteogenic differentiation capacity (Fig 5G and H), suggesting that D292E osteoclasts may exacerbate inhibition of osteogenesis through paracrine signaling.

### *SLC2A10* variant drives glycolysis-dependent macrophage polarization and osteoclastogenesis

To further elucidate the molecular basis underlying these cellular phenotypes, we next explored the associated signaling and metabolic mechanisms. GLUT10 acts as a transmembrane protein that primarily transports glucose and regulates intracellular glucose metabolism. Macrophage polarization is closely associated with metabolic reprogramming, particularly of glucose metabolism. Glycolysis is reprogrammed to facilitate and sustain the M1 inflammatory phenotype. Furthermore, increased glycolytic activity has been positively correlated with monocyte/macrophage activation and osteoclast differentiation. Therefore, we aimed to investigate whether *SLC2A10* variants influence glycolytic levels, thereby altering the metabolic state within macrophages.

RT–qPCR analysis revealed that in LPS-induced M1-polarized RAW264.7 macrophages (24-h induction), the D292E group displayed significantly higher mRNA expression of key glycolytic rate-limiting enzymes compared with the WT group, indicating

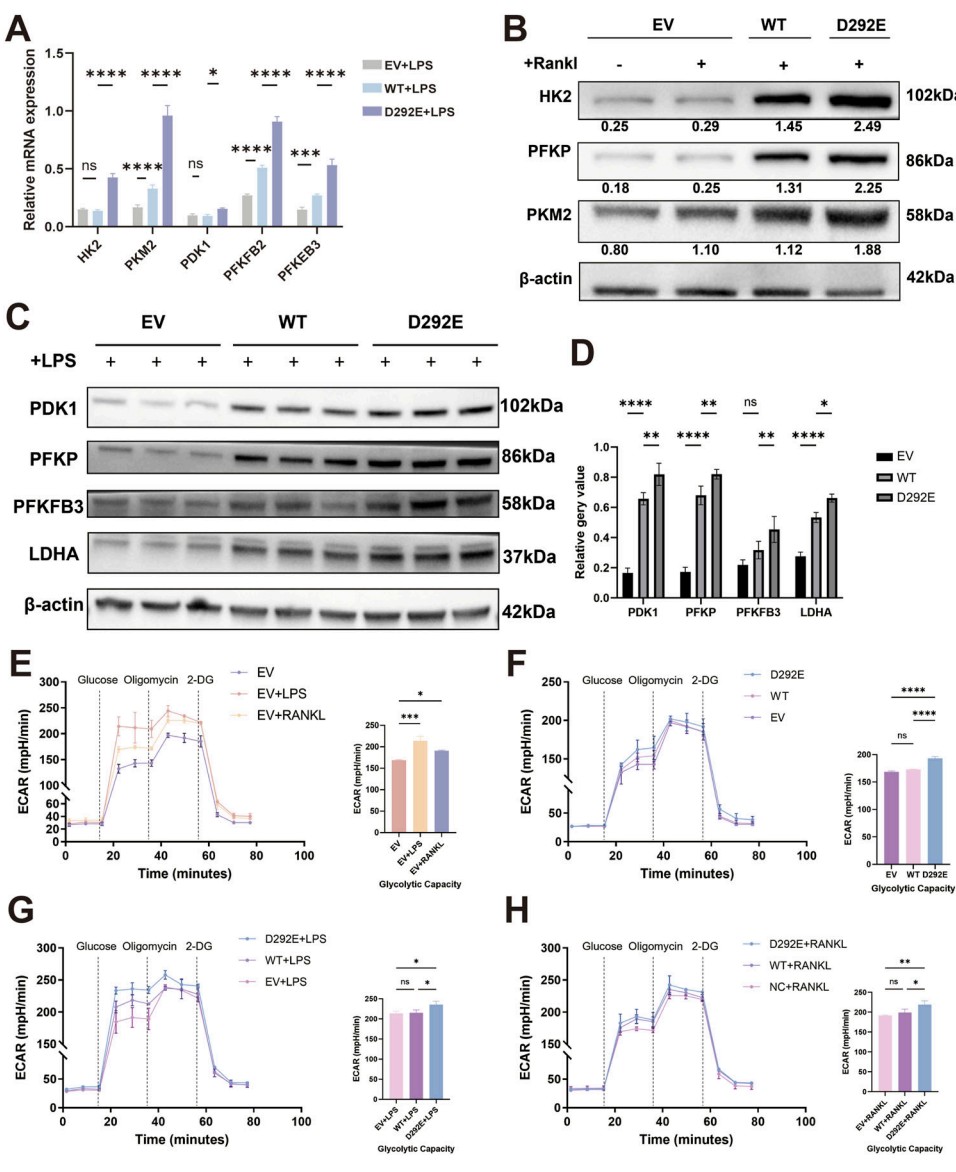

**Figure 6. Monitoring of cellular glucose metabolism in M1-type macrophages and osteoclasts.**

**(A)** mRNA expression of glycolytic markers in EV, WT, and D292E groups at 24 h after LPS stimulation. **(B)** Protein expression of glycolytic markers at 5 d after RANKL induction. **(C)** Protein expression of glycolytic markers at 24 h after LPS stimulation. **(C, D)** Quantification of results in (C). **(E)** Glycolytic function assay in EV, EV + LPS, and EV + RANKL groups. **(F)** Glycolytic function assay in EV, WT, and D292E groups. **(G)** Glycolytic function assay in EV, WT, and D292E groups at 24 h after LPS stimulation. **(H)** Glycolytic function assay in EV, WT, and D292E groups at 2 d after RANKL induction. EV: empty vector cells; WT: overexpression of WT *SLC2A10* cells; D292E: overexpression of variant *SLC2A10* cells. N = 3 per group. One-way ANOVA was used to evaluate the significance between multiple groups, and a t test was used to calculate the *P*-value for post hoc comparisons. All data are shown as the mean ± SD; *P < 0.05, **P < 0.01, ***P < 0.001, ****P < 0.0001.

enhanced glycolytic activity (Fig 6A). Western blot further demonstrated that (1) under LPS stimulation, the D292E group exhibited markedly increased protein expression of glycolytic enzymes HK2 and PKM2 compared with the WT group (Fig 6C and D); and (2) after RANKL induction, glycolytic enzyme expression in the D292E group was also significantly elevated relative to both the WT and Vector groups (Fig 6B). Enhanced glucose consumption together with elevated acid output in D292E cells suggests a metabolic shift toward augmented glycolysis (Fig S2C). These results suggest that the variant broadly increases macrophage metabolic dependence on glycolysis.

Seahorse real-time metabolic analysis showed that either LPS or RANKL stimulation significantly augmented the glycolytic capacity of RAW264.7 cells (Fig 6E), suggesting that glycolytic reprogramming may coordinately regulate both M1 macrophage polarization and osteoclast differentiation.

Metabolic profiling of Vector, WT, and D292E groups under different treatment conditions provided consistent evidence: the D292E group maintained elevated glycolytic activity across all conditions—at resting state, after 1 d of LPS induction, and after 2 d of RANKL induction (Fig 6F–H). This persistent glycolytic advantage underscores the distinctive metabolic phenotype conferred by the variant.

### The *SLC2A10* D292E variant increases the inflammation level and disrupts bone homeostasis in zebrafish

To validate the effects of the *SLC2A10* variant in vivo, we established a zebrafish model. Zebrafish embryos were microinjected with mRNA encoding WT *SLC2A10* or the c.876C>A (p.D292E) variant to investigate its impact on skeletal development and inflammatory regulation. RT–qPCR analysis revealed that at

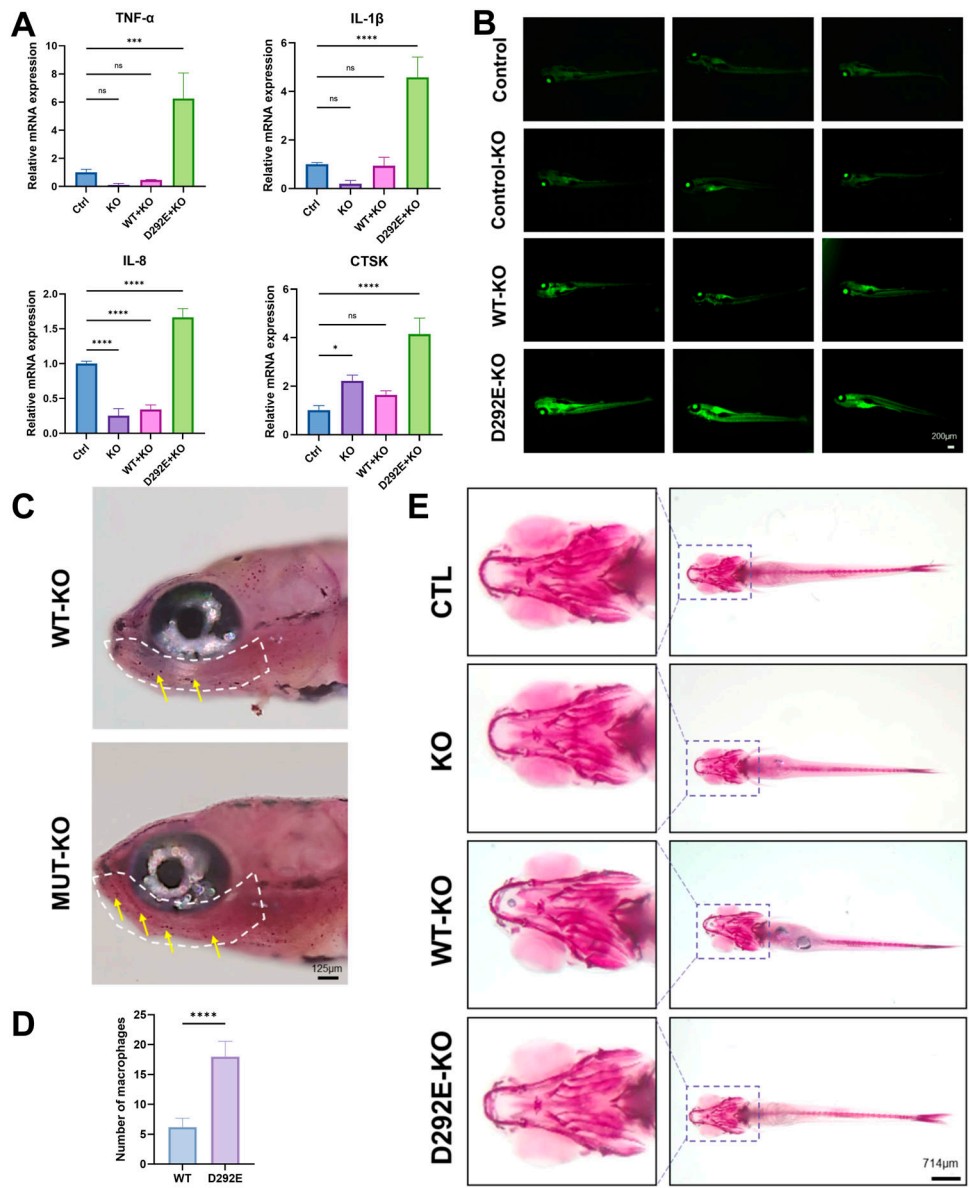

**Figure 7. Zebrafish model injected with *SLC2A10* mRNA and mutated mRNA.**
**(A)** mRNA expression of inflammatory and osteoclast markers at 4 d postfertilization in four groups. **(B)** Quantification of ROS fluorescence intensity in control, KO, WT, and D292E groups. **(C)** Representative neutral red staining images of 12-dpf zebrafish larvae showing neutral red–positive cells/signals in the mandibular region. The white dashed line delineates the jawbone region, and yellow arrows indicate representative neutral red–positive cells. **(D)** Quantification of neutral red–positive cells/signals within the mandibular region of interest. **(E)** Representative Alizarin Red staining images of 21-dpf zebrafish larvae showing skeletal mineralization patterns. CTL: healthy control zebrafish; KO: *SLC2A10* knockout zebrafish; WT: zebrafish model injected with *SLC2A10* mRNA; D292E: zebrafish model injected with mutated mRNA. N = 10 per group. One-way ANOVA was used to evaluate the significance between multiple groups, and a *t* test was used to calculate the *P*-value for post hoc comparisons. All data are shown as the mean ± SD; *$P < 0.05$, **$P < 0.01$, **$P < 0.001$, ****$P < 0.0001$.

4 d postfertilization (dpf), *SLC2A10* D292E–expressing zebrafish exhibited significantly elevated mRNA levels of inflammatory factors (TNF-α, IL-1β) and osteoclast markers (CTSK) compared with WT controls (Fig 7A).

DCFH-DA fluorescent probe detection in 12 dpf larvae demonstrated markedly increased ROS accumulation in the D292E group (Fig 7B), indicating impaired ROS scavenging capacity. Neutral red staining showed an increased accumulation of neutral red–positive cells/signals in D292E larvae, with a more pronounced signal in the mandibular region (Fig 7C). Because neutral red staining is not a lineage-specific macrophage marker, these findings were interpreted as an increase in neutral red–positive phagocytic activity/cells rather than direct evidence of macrophage expansion. Quantification of neutral

red–positive cells/signals within the mandibular region of interest further supported this increase in the D292E group (Fig 7D).

Representative Alizarin Red staining images of 21 dpf larvae showed a trend toward reduced calcified area in the D292E group compared with controls (Fig 7E). Although these images are consistent with a potential effect of the *SLC2A10* variant on skeletal mineralization, they should be interpreted cautiously because no quantitative analysis of Alizarin Red staining was performed.

Collectively, these findings demonstrate that the *SLC2A10* variant promotes bone metabolic imbalance through enhanced inflammatory responses, aggravated oxidative stress, aberrant macrophage recruitment, and impaired bone formation.

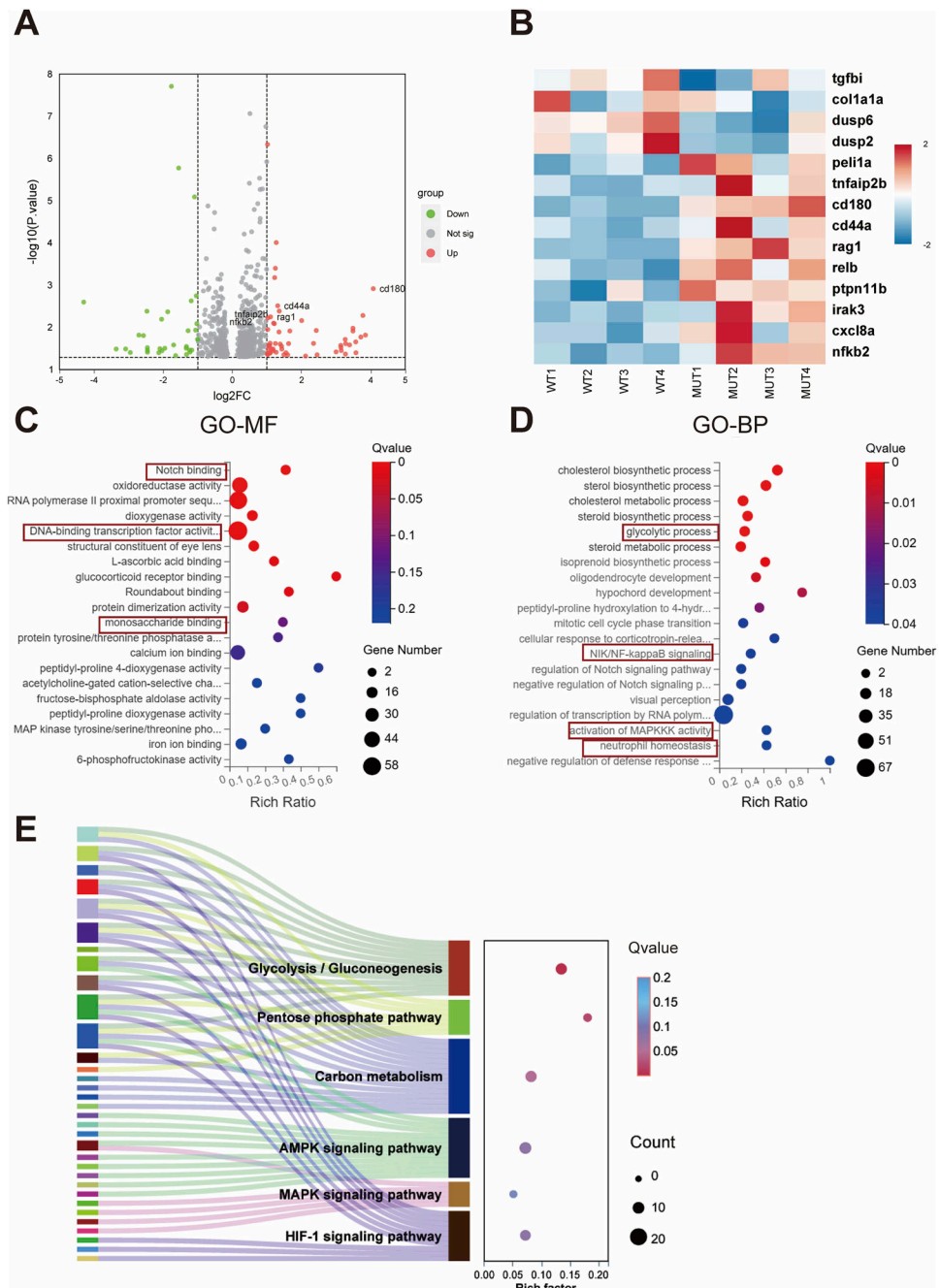

**Figure 8. Gene expression analysis by RNA-seq performed on zebrafish injected with *SLC2A10* mRNA and mutated mRNA at 4 dpf.**
**(A)** Volcano plot of differentially expressed genes. **(B)** Heatmap depicting the expression pattern of differentially expressed genes. **(C)** Dotplot for Gene Ontology–term enrichment analysis including top 20 molecular function, compared between MUT and WT. **(D)** Dotplot for Gene Ontology–term enrichment analysis including top 20 biological process, compared between MUT and WT. **(E)** Sankey bubble plot of KEGG pathway enrichment (top 6 pathways) in D292E variant versus WT groups. WT: zebrafish model injected with *SLC2A10* mRNA; MUT: zebrafish model injected with mutated mRNA. *P*-values: *$P < 0.05$, **$P < 0.01$, ***$P < 0.001$; ns: no significant difference.

## Transcriptomic analysis reveals glycometabolism–inflammation–osteoclastogenesis network

To assess the impact of the *SLC2A10* variant on global transcription, mRNA sequencing was conducted in WT and D292E zebrafish at 4 dpf. Transcriptomic sequencing analysis elucidated the systemic impact of the *SLC2A10* variant on inflammatory, metabolic, and osteogenic regulatory networks. DEG analysis revealed that the volcano plot showed significant up-regulation of inflammatory markers, such as CD180 and CD44a,

in the D292E group (Fig 8A). The heatmap further confirmed a characteristic pro-inflammatory signature, with elevated pro-inflammatory factors accompanied by down-regulation of anti-inflammatory factors, including TGFBI and members of the dusp gene family (Fig 8B).

Functional annotation analysis demonstrated that GO-MF terms were significantly enriched for processes related to the Notch pathway (osteoclast differentiation regulation), DNA transcriptional regulation (inflammation-related), and monosaccharide binding (glycometabolism) (Fig 8C). GO-BP analysis indicated

strong enrichment in glycolysis, NF-κB signaling, MAPK cascade, and neutrophil chemotaxis (Fig 8D). Pathway enrichment analysis, illustrated by a Sankey diagram, revealed significant activation of glycometabolic pathways (glycolysis/gluconeogenesis, carbon metabolism) in the D292E group, alongside concurrent enrichment of inflammatory pathways (AMPK, MAPK) and immune response pathways (Fig 8E).

Collectively, these multi-omics data indicate that the *SLC2A10* variant drives molecular cascades of bone metabolic imbalance through the synergistic activation of glycolytic reprogramming, pro-inflammatory signaling networks, and osteoclast differentiation regulatory pathways.

## Discussion

In this study, we identify a previously unreported *SLC2A10* missense variant (c.876C>A, p.Asp292Glu) associated with mandibular CNO. Functional analyses demonstrate that this gain-of-function variant enhances glycolytic reprogramming in macrophages, promotes polarization toward a pro-inflammatory M1 phenotype, and accelerates osteoclast differentiation. These findings suggest a potential link between GLUT10-associated metabolic regulation, immunometabolic activation, and inflammatory bone destruction, although this proposed mechanistic framework requires further validation.

GLUT10, encoded by *SLC2A10*, is a conserved transmembrane glucose transporter comprising 12 predicted transmembrane domains (Dawson et al, 2001; Mueckler & Thorens, 2013). The identified D292E substitution lies within a transmembrane region and may alter hydrogen bonding, polarity, or transporter dynamics, potentially affecting glucose transport efficiency. Mutations in *SLC2A10* have been predominantly associated with arterial tortuosity syndrome, where they are typically inherited in an autosomal recessive manner and lead to loss of function of the encoded protein (Ritelli et al, 2014; Boel et al, 2020). In contrast, the variant identified in our study is a heterozygous dominant mutation, and functional analyses indicate that it confers a gain-of-function effect (Fig S2A and B). These findings suggest that distinct mutation types and loci within *SLC2A10* may differentially alter protein activity, ultimately giving rise to divergent pathological manifestations.

Macrophages are central mediators of inflammatory signaling in autoimmune and autoinflammatory diseases (Hamidzadeh et al, 2017). Activation of pattern recognition receptors triggers kinase-dependent pathways converging on transcription factors including NF-κB, CREB, and interferon regulatory factors, thereby initiating the transcription of pro-inflammatory cytokines (Medzhitov & Horng, 2009). Among these cytokines, TNF functions as a key amplifier of inflammatory cascades by promoting IL-6, IL-12/23(p40), and type I interferon expression (Janeway & Medzhitov, 2002; Rosenzweig & Holland, 2005). In addition to orchestrating inflammatory responses, macrophages also play a critical role in regulating bone homeostasis and inflammatory bone diseases. Consistent with this role, genetic studies have implicated macrophage dysregulation in CNO. For example, variants in P2RX7 have been shown to alter inflammasome assembly and cytokine release

in affected individuals (Charras et al, 2024). In line with these observations, our results demonstrate that macrophages carrying the mutation exhibit increased expression of pro-inflammatory cytokines, suggesting enhanced inflammatory activation that may contribute to disease pathogenesis (Fig 4A–G).

Macrophage activation is tightly coupled to metabolic remodeling (Liu et al, 2021). Glycolytic reprogramming, characterized by a shift from oxidative phosphorylation to aerobic glycolysis, is a hallmark of inflammatory M1 macrophages (Tannahill et al, 2013; Mills et al, 2016). Increased glucose uptake and lactate production, reminiscent of the Warburg effect (Russell et al, 2019; Liu et al, 2021), generate metabolic intermediates and ROS that reinforce inflammatory signaling (Mahon et al, 2020; Palsson-McDermott & O'Neill, 2013; Diskin & Pålsson-McDermott, 2018). Our data indicate that the D292E variant enhances glucose uptake and glycolytic flux, suggesting that aberrant GLUT10 activity directly promotes pro-inflammatory metabolic states (Fig 6A–H).

Osteoclasts arise from monocyte/macrophage precursors and are critically dependent on M-CSF and RANKL signaling (Udagawa et al, 1990; Dougall et al, 1999; Boyle et al, 2003; Koide et al, 2013). Upon RANK activation, downstream pathways including NF-κB and ERK are triggered, which are accompanied by profound metabolic remodeling during osteoclast differentiation (Park-Min, 2019; Liu et al, 2023). In this context, metabolic reprogramming toward glycolysis has emerged as a key requirement for osteoclastogenesis. Up-regulation of glycolytic regulators such as GLUT1 and LDHA is essential for osteoclast differentiation, and disruption of these pathways impairs osteoclastogenesis (Indo et al, 2013; Ahn et al, 2016; Li et al, 2020). Consistently, osteoclast resorptive activity is likewise linked to aerobic glycolysis and lactate production (Indo et al, 2013; Arnett & Orriss, 2018; Taubmann et al, 2020). Notably, genetic disorders affecting metabolic regulators further support this concept. Mutations in *LPIN2*, which cause Majeed syndrome, similarly enhance osteoclast formation and inflammatory bone lesions (Bhuyan et al, 2021), reinforcing the concept that metabolic dysregulation within the macrophage–osteoclast lineage contributes to autoinflammatory bone pathology. Consistent with this concept, our results demonstrate that mutant macrophages exhibit enhanced osteoclast differentiation accompanied by increased glycolytic activity (Figs 5A–I and 6B and H).

Currently, treatment for CNO is largely based on symptomatic control, with NSAIDs as first-line therapy and corticosteroids, immunosuppressants, bisphosphonates, or TNF-α inhibitors used in refractory cases (Schnabel et al, 2017; Girschick et al, 2018; Buch et al, 2019; Concha et al, 2020). However, the pathophysiology of CNO remains incompletely understood and high-quality clinical evidence is limited, posing challenges for optimal management (Roberts et al, 2024). In this study, we identify metabolic reprogramming driven by the *SLC2A10* variant, characterized by enhanced glycolysis, as a potential pathogenic pathway. Notably, pharmacological inhibition of glycolysis with 2-deoxy-D-glucose (2-DG) partially rescued the abnormal cellular phenotype in vitro, suggesting that metabolic targeting may represent a potential therapeutic strategy.

Several limitations should be acknowledged. This study is based on a single family, limiting conclusions regarding population-level prevalence or penetrance. Although our data indicate altered

metabolic activity in D292E-expressing macrophages, the precise contribution of GLUT10 localization and ascorbate transport requires further investigation. The zebrafish staining assays also require cautious interpretation, as neutral red staining is not a lineage-specific macrophage marker and the Alizarin Red staining observations were based on representative images rather than quantitative analysis. Future studies incorporating multicenter cohorts and CRISPR/Cas9-mediated knock-in models in zebrafish or mice, including tissue-specific systems, will be necessary to validate GLUT10 function in vivo and clarify its contribution to chronic bone immune dysregulation.

In summary, our findings position GLUT10-mediated glycolytic reprogramming as a potential upstream driver of macrophage activation and osteoclast-mediated bone destruction in mandibular CNO. By integrating genetic discovery with immunometabolic mechanisms, this study expands the conceptual framework of autoinflammatory bone disease and highlights metabolic regulation as a potential therapeutic axis.

# Materials and Methods

### Genetic and segregation analysis of the family

Whole-exome sequencing for the family was performed by Novogene Co., Ltd. Quality-filtered reads were aligned to the human reference genome (GRCh38) using BWA. Variant filtering was conducted using a sequential strategy. Variants with a minor allele frequency >1% in population databases (1000 Genomes, ESP6500, gnomAD ALL/EAS) were excluded. The analysis was restricted to exonic and splice-site variants. Nonconserved synonymous variants and short insertions/deletions (<10 bp) located in repeat regions were excluded unless predicted to affect splicing.

Candidate variants were further evaluated under an autosomal dominant inheritance model. Variants predicted to be damaging by in silico tools (CADD, SIFT, PolyPhen-2, MutationTaster) or predicted to affect splicing (dbscSNV) were prioritized for further analysis. All retained variants were classified according to the American College of Medical Genetics and Genomics (ACMG) guidelines. Segregation analysis within the family was performed to assess cosegregation with the disease phenotype.

After filtering and segregation analysis, two candidate variants were identified: *ALK* p.R311H and *SLC2A10* p.D292E. The ALK variant was excluded because its distribution among tested family members was not consistent with the disease phenotype, whereas the *SLC2A10* variant was detected in all tested affected individuals and was absent in the tested unaffected relatives. These findings support segregation of the *SLC2A10* variant with the disease phenotype within the genotyped members of the family.

Genomic DNA was extracted from peripheral blood samples of family members using the Magi D3018-03 kit (Magen). Sanger sequencing was performed on an Applied Biosystems platform using primers listed in Table S3. Sequencing chromatograms were analyzed using SnapGene version 6.0.2.

### Structural modeling and molecular simulation analysis of *SLC2A10* WT and variant

The three-dimensional structure of the human *SLC2A10* WT protein was predicted using the AlphaFold protein structure database (https://alphafold.com/) (Fig 1). The structural model of the *SLC2A10* p.D292E variant was generated using the Swiss-Model protein structure database (https://swissmodel.expasy.org/interactive), with O95528.1.A as the template (Fig 1).

### Protein microarray analysis

Human protein antibody arrays (Quantibody Human Inflammation Array 3, Cat. # GSH-INF-3; RayBiotech, Inc.) were applied following the supplier's protocol. Each slide included both negative and positive controls. The positive controls consisted of defined quantities of biotinylated IgGs printed directly on the membrane. Fluorescence signals were quantified by subtracting local background and subsequently normalized to the positive control values. The expression of 40 cytokines was compared between normal control and CNO samples, and the statistical significance of differences was assessed. The expression ratio indicated the direction and magnitude of changes in protein expression.

Heatmap visualization of cytokine expression was performed using normalized fluorescence intensity values. Hierarchical clustering was conducted to illustrate expression patterns between control and CNO samples. Data visualization and clustering were performed using R software. For functional interpretation, differentially expressed cytokines were mapped to Kyoto Encyclopedia of Genes and Genomes (KEGG) pathways using the KEGG database for annotation purposes. Given that the array represents a predefined inflammation-focused panel, no statistical pathway enrichment analysis was performed.

### Plasmid construction

Lentiviruses encoding mouse *Slc2a10* WT and *Slc2a10* (D292E) mutant sequences were purchased from GeneChem Inc. The lentiviral vector GV492 (pGC-FU-3FLAG-CBh-gcGFP-IRES-puromycin) was digested with AgeI and BamHI restriction enzymes, and the *Slc2a10* WT or mutant sequences were inserted using the In-Fusion recombination method. Recombinant vectors were verified by DNA sequencing.

The WT *Slc2a10* template sequence was obtained from the GeneChem library and amplified using specific primers. The *Slc2a10* D292E mutant was generated by site-directed mutagenesis PCR targeting the variant site. The resulting PCR products were mixed in equal amounts, recombined in *Escherichia coli*, and subjected to plasmid extraction and verification. Single bacterial clones were selected for sequencing, which confirmed that the expression vectors carried the intended site-directed variant. Sequences of primers used for mutagenesis are listed in Table S2.

### Virus production

Recombinant lentiviruses encoding either WT *SLC2A10* or the c.876C>A (p.D292E) variant, together with psPAX2 and pMD2.G

helper plasmids, were cotransfected into 293T cells using Invitrogen Lipofectamine 2000 (Cat. #11668-030; Thermo Fisher Scientific) following the supplier's instructions. At 72 h post-transfection, viral supernatants were collected, centrifuged to clear cell debris, and filtered using 0.45-$\mu$m cellulose acetate filters (Millipore). The viral titer, estimated by flow cytometry detection of GFP-positive 293T cells, was ~5 × 10$^8$ TU/ml. Prepared aliquots were stored at −80°C until further application.

## Cell cultures and transfection

RAW264.7 mouse macrophages were maintained in DMEM (Thermo Fisher Scientific) supplemented with 10% FBS (Sigma-Aldrich) at 37°C under a humidified 5% $CO_2$ atmosphere. Cells were plated in six-well dishes and allowed to adhere for 24 h before transduction with lentiviruses carrying either WT *Slc2a10* or the p.D292E variant. Transductions were performed using HiTrans G lentiviral transduction reagent (GeneChem) with an MOI of 10. After 24 h, cells were subjected to puromycin selection to establish stable transductants. At 48 h post-transduction, more than 90% of cells exhibited GFP fluorescence under a fluorescence microscope, confirming successful transduction. The transduced cells were subsequently maintained in complete medium for downstream experiments.

## Western blot

Cells and tissue specimens were rinsed with ice-cold PBS and lysed using RIPA buffer (Huaxingbio) containing phosphatase and protease inhibitor cocktails. Cell lysates were clarified by centrifugation at 12,000$g$ for 15 min at 4°C, and protein levels were measured using a BCA protein assay kit (Thermo Fisher Scientific). Equivalent amounts of protein were subjected to 10% SDS–PAGE (NCM Biotech) and transferred onto 0.22-$\mu$m PVDF membranes (Millipore) using a Trans-Blot apparatus (Bio-Rad). The membranes were incubated in TBST containing 5% skim milk for 1 h to block nonspecific binding at room temperature and subsequently incubated overnight at 4°C with primary antibodies (listed in Table S5). After washing, HRP-conjugated secondary antibodies were applied for 1 h at 25°C. Protein bands were detected using enhanced chemiluminescence reagents (Millipore) and visualized using a ChampChemi gel imaging system (Sagercreation). Antibody information is provided in Table S5.

## Zebrafish model

AB strain WT zebrafish were purchased from the China Zebrafish Resource Center (Wuhan, China) and housed in a circulating aquaculture system at 28.5°C under a 14-h light/10-h dark cycle. Experimental procedures were approved by the Animal Care and Use Committee of Peking University (Approval No. DLASBE0528). To generate the constructs, coding sequences of *slc2a10* WT and D292E mutant were subcloned into the pcDNA3.1 vector. The recombinant plasmids were transformed into *E. coli*, amplified, purified, and verified by sequencing. The confirmed plasmids were then used as templates for in vitro transcription. mRNAs were synthesized using the mMESSAGE mMACHINE T7 kit (Thermo Fisher

Scientific) according to the manufacturer's protocol, purified, diluted to 100 ng/$\mu$l, and injected (1 nl) into one-cell-stage embryos using a microinjection system. The zebrafish embryo mRNA injection concentration test is shown in Table S6.

Embryos were cultured in E3 solution composed of 5 mM NaCl, 0.33 mM $MgSO_4$, 0.33 mM $CaCl_2$, and 0.17 mM KCl at 28.5°C until analysis. Phenotypic evaluations included morphological assessment, histological staining, and molecular analyses as described in subsequent sections.

## Morphological and histological analyses

Fresh mandibular tissue samples from patients were fixed in 4% PFA for 24 h at room temperature, dehydrated, and paraffin-embedded. Serial sections of 4-$\mu$m thickness were stained with hematoxylin and eosin (H&E) for histological examination or with a TRAP kit (Cat. #294-67001; FUJIFILM Wako) for osteoclast detection. Images were acquired using an Olympus BX53 microscope (Olympus) and quantified with Image-Pro Plus 6.0 software.

For zebrafish skeletal analysis, larvae were fixed in 4% PFA, followed by Alizarin Red staining (Beyotime) to visualize calcified structures. Images were obtained using a Zeiss AxioZoom.V16 stereo microscope (Carl Zeiss). ROS in zebrafish larvae at 12 d post-fertilization were detected using 1 $\mu$M DCFH-DA (Beyotime) staining. Fluorescence images were acquired with an Olympus IX73 inverted microscope and quantified using Image-Pro Plus software (Olympus).

## TRAP staining and osteoclast quantification in RAW264.7 cells

RAW264.7 cells were seeded in six-well plates and stimulated with RANKL (50 ng/ml) for 10 d to induce osteoclast differentiation. Cells were then fixed with 4% PFA and stained using a TRAP kit (Cat. #294-67001; FUJIFILM Wako) according to the manufacturer's instructions.

Three independent wells per group were analyzed. Each well was divided into nine predefined regions, and images were acquired under identical magnification settings. TRAP-positive multinucleated cells (≥2 nuclei) were manually counted in all regions. The total number of multinucleated cells per well was calculated, and the mean of three wells was used for statistical analysis.

## Seahorse metabolic analysis

Extracellular acidification rate was assessed using a Seahorse XF96 extracellular flux analyzer (Agilent Technologies) following the manufacturer's instructions. RAW264.7 cells were plated at 6,000 cells per well in XF96 culture plates and cultured overnight before analysis. For extracellular acidification rate measurement, after specific stimulation, the medium was replaced with Seahorse basal medium (1 mM pyruvate, 2 mM glutamine, and 10 mM glucose). The cells were then cultured for 1 h at 37°C in a $CO_2$-free incubator. Finally, the plate was sequentially injected with glucose (10 mM), oligomycin (1 $\mu$M), and 2-DG (50 mM) from the XF Cell Mito Stress Test Kit (Agilent Technologies).

## RNA extraction, RT–PCR, and RNA sequencing

Total RNA was isolated from cultured cells with TRIzol reagent (Invitrogen) in accordance with the manufacturer's instructions. cDNA synthesis was performed using the PrimeScript RT Master Mix. Quantitative PCRs (qPCRs) were prepared with cDNA, SYBR Green Master Mix, primers, and nuclease-free water, and amplification was carried out on the Archimed real-time PCR system (RocGene). The sequences of primers used are listed in Table S4.

For RNA sequencing, mRNA was enriched from total RNA with oligo(dT)-conjugated magnetic beads. The purified mRNA was fragmented, reverse-transcribed into cDNA, end-repaired, A-tailed, and ligated to sequencing adapters. The resulting cDNA libraries were PCR-amplified and sequenced on the DNBSEQ platform (BGI-Shenzhen). Raw sequencing reads were processed to eliminate adapter sequences, low-quality reads, and reads containing ambiguous bases. Filtered high-quality reads were mapped to the reference genome using HISAT2, and transcript abundance was estimated with RSEM. Differentially expressed genes were identified using DESeq2, applying a threshold of Q ≤ 0.05. Functional enrichment of Gene Ontology (GO) categories and KEGG pathways was assessed by hypergeometric testing with multiple testing correction.

## Flow cytometry and ROS detection

To analyze cell populations, $1 \times 10^6$ RAW264.7 cells were collected after specific treatments and stained with various dyes according to the manufacturer's protocols. Stained cells were washed three times with the appropriate buffer, centrifuged (at $60g$ for 5 min at 25°C), and resuspended in 500 $\mu$l of buffer. The dyes used included CD86 (Cat. #AC0047; Beyotime), F4/80 (Cat. #AC1328; Beyotime), and dihydroethidium (Cat. #S0063; Beyotime) for intracellular ROS detection. Samples were analyzed immediately on a BD flow cytometer (BD Biosciences), and data were processed using FlowJo software version 10.8.1. Mean fluorescence intensity values were calculated from three independent biological replicates for statistical analysis.

For ROS detection in zebrafish, 12 dpf larvae were anesthetized with tricaine and stained with 1 $\mu$M DCFH-DA (Cat. #S0034S; Beyotime) to evaluate ROS levels. After washing, larvae were imaged using an Olympus IX73 fluorescence microscope under identical exposure settings for all groups. For each experimental condition, at least 10 larvae were analyzed per replicate. Fluorescence intensity was quantified using ImageJ software (NIH) after background subtraction. For each larva, the mean fluorescence intensity within the defined region of interest was measured. Three independent experiments were performed, and averaged values were used for statistical analysis.

## Knockout plasmids

The plasmid expressing short guide RNA (sgRNA) targeting the sequence of the *SLC2A10* gene (CTCCTCGCCTCCCTTGTCGG) and negative control (CGCTTCCGCGGCCCGTTCAA) were synthesized and cloned into GV708 (U6-sgRNA-EF1a-spCas9-FLAG-CMV-EGFP-P2A-

puro) vector with BsmBI sites (purchased from Shanghai Genechem Co., Ltd.); the recombinant vector was detected by DNA sequencing. The final products were then transfected into *E. coli*. DH5α followed by extraction with the Endofree plasmid Mega kit (QIAGEN) obtained the sgRNA-*SLC2A10* plasmid.

## Transfection of RAW264.7 cells

RAW264.7 macrophage cells were seeded in six-well plates and cultured to ~70–90% confluence before transfection. Plasmid transfection was performed using Lipofectamine 3000 (Thermo Fisher Scientific) according to the manufacturer's instructions.

For each well of a six-well plate, 2.5 $\mu$g of plasmid DNA was diluted in 125 $\mu$l of Opti-MEM and mixed with 5 $\mu$l of P3000 reagent. In a separate tube, 7.5 $\mu$l of Lipofectamine 3000 reagent was diluted in 125 $\mu$l of Opti-MEM. The diluted DNA solution was then combined with the diluted Lipofectamine 3000 reagent at a 1:1 ratio and incubated at room temperature for 10–15 min to allow DNA–lipid complex formation.

The DNA–lipid complexes were subsequently added to the cells and incubated at 37°C in a humidified 5% $CO_2$ atmosphere. Transfected cells were analyzed 24–48 h post-transfection.

## Glucose uptake assay

Cellular glucose uptake was measured using a commercial glucose uptake assay kit (Cat. #S0556S; Beyotime) according to the manufacturer's instructions. This assay is based on the uptake of 2-deoxy-D-glucose (2-DG) by cells and its phosphorylation to 2-DG-6-phosphate (2-DG6P). Through coupled enzymatic reactions, $NADP^+$ is converted to NADPH, which subsequently reduces oxidized glutathione via a glutathione reductase system. The reduced glutathione reacts with DTNB to generate the yellow product TNB, and absorbance at 412 nm is measured to indirectly quantify glucose uptake.

RAW264.7 cells were seeded in 96-well plates ($1 \times 10^4$ cells per well). After experimental treatments, cells were washed with PBS and incubated overnight in serum-free low-glucose medium for starvation. The next day, cells were equilibrated in KRPH buffer at 37°C for 40 min to deplete residual glucose. Cells were then incubated with 10 mM 2-DG for 20 min. Where indicated, insulin (100 $\mu$g/ml) was used as a positive control.

After incubation, cells were washed and lysed with glucose uptake lysis buffer. Cell lysates were centrifuged at $14,000g$ for 5 min at 4°C, and supernatants were collected for analysis. A standard curve was generated using serial dilutions of 2-DG6P (0–100 pmol). Samples and standards were subjected to sequential enzymatic reactions according to the kit protocol, and absorbance at 412 nm was measured using a microplate reader.

Glucose uptake was calculated based on the standard curve and normalized after subtraction of background control values.

## Human sample collection

Peripheral blood samples were collected from participating individuals using standard venipuncture procedures. Genomic DNA was extracted from fresh whole blood and subjected to sequencing

## Life Science Alliance

by Novogene. Mandibular lesion tissues were obtained from CNO patients during clinically indicated biopsy procedures. Control mandibular bone tissues were collected from age-matched individuals undergoing orthognathic surgery without inflammatory or metabolic bone disease.

All sample collection procedures were performed in accordance with institutional ethical guidelines and after obtaining informed consent from all participants or their legal guardians.

### Statistical analysis

Data are reported as the mean value with corresponding SEM. Statistical analyses were conducted using GraphPad Prism v10.0.0 or R v4.5.0. The Shapiro–Wilk test was used to assess normality. For comparisons between two groups, normally distributed data were analyzed with unpaired two-tailed $t$ tests, whereas non-normally distributed data were assessed using Mann–Whitney tests. For multiple group comparisons, one-way or two-way analysis of variance (ANOVA) was conducted, followed by Dunnett's or Tukey's post hoc tests, respectively. A $P$-value < 0.05 was considered statistically significant.

## Data Availability

Because of restrictions from the patient consent approved by the research ethics committees, it is not possible to deposit complete exome sequencing data in a public repository, but the data could be made available to interested researchers by contacting the corresponding author (J An). RNA-seq data have been deposited in NCBI database under BioProject accession number PRJNA1446281. The source data of this article are collected in the following database: https://www.ebi.ac.uk/biostudies/studies/S-BSST2886.

### Study approval

Written informed consent was obtained from all subjects and family members. The study was approved by the Biomedical Ethics Committee of Peking University School and Hospital of Stomatology (Approval No. PKUSSIRB-202272025) and performed in accordance with the ethical standards laid down in the 1964 Declaration of Helsinki.

All animal experiments were approved by the Experimental Animal Ethics Committee of Peking University Health Science Center (Animal Protocol No. DLASBE0528) and conducted in accordance with institutional guidelines for the care and use of laboratory animals.

For additional methods and details, see Tables S1, S2, S3, S4, S5, and S6.

## Supplementary Information

## Acknowledgements

This study was supported by the National Science and Technology Department Project (Nos.2020YFF0305104).

### Author Contributions

X Li: conceptualization, software, validation, investigation, visualization, methodology, and writing—original draft.
L Shen: software and validation.
K Jia: supervision, investigation, project administration, and writing—review and editing.
S Chen: data curation, supervision, project administration, and writing—review and editing.
J An: resources, software, formal analysis, supervision, funding acquisition, project administration, and writing—review and editing.

### Conflict of Interest Statement

The authors declare that they have no conflict of interest.

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
