## [Reviewer comments · Life Science Alliance]

A SLC2A10 Gain-of-Function Variant Links Macrophage Glycolysis to Chronic Nonbacterial Osteomyelitis

Xiang Li, Li Shen, Kuan Jia, Shuo Chen, and Jingang An

DOI: <https://doi.org/10.26508/lsa.202603772>

Corresponding author(s): Jingang An, Peking University Stomatological Hospital; Jingang An, Peking University Stomatological Hospital; and Shuo Chen, Peking University School and Hospital of Stomatology

Review Timeline:

Submission Date:	2026-05-14
Editorial Decision:	2026-05-18
Revision Received:	2026-05-24
Accepted:	2026-05-27

Scientific Editor: Tim Fessenden

Transaction Report:

Please note that the manuscript was previously reviewed at another journal and the reports were taken into account in the decision-making process at *Life Science Alliance*.

Referee #4 Review

Remarks for Author:

In their revised manuscript the authors have attempted to address many critical comments raised by three reviewers. Although some issues have now been clarified, which also included additional experiments, there are many remaining problems, which question the relevance of the respective findings.

One remaining problem is the limited genetic evidence allowing to conclude that the p.Asp292Glu variant of SLC2A10 is indeed causative for the CNO manifestation. Since this variant has a relatively high frequency (0.0009359 in East Asia, according the gnomAD), and since apparently only one unaffected family member (II-2) was genetically tested, it is still an issue that the conclusions are based on only one family.

In their response letter, the authors state that by screening 37 additional CNO patients they identified another variant in SLC2A10, which will be investigated in future studies. Given the fact that inactivating variants in SLC2A10 are known to cause arterial tortuosity syndrome, this putative gain-of-function variant should be tested, similar to a known loss-of-function variant causing arterial tortuosity syndrome, as suggested by reviewer #2 (comment #7).

With respect to other molecular experiments, the authors tried to address most of the critical comments, for instance by using a glycolysis inhibitor. However, although these changes surely improved the quality of the manuscript, there are still remaining issues, for instance the lack of quantification for the Western Blot analysis shown in Figure EV2.

Referee #5 Review

Comments on Novelty/Model System for Author:

The zebrafish model is not adequate to examine the involvement of SLC2A10 in CNO. The jaw bone histology to see macrophages, osteoclasts and osteoblasts could not be performed to evaluate osteomyelitis. As the clinical evidence is weak as it is based on 4 WES of only one family, a more rigorous animal model adequate to assess CNO is necessary.

Remarks for Author:

The potential involvement of SLC2A10 variant in CNO is interesting. However, the clinical evidence is not sufficient as the sequencing analysis was performed on only one family and the information on clinical evaluation of the family members is limited. To gain firm evidence on the causal relationship between the SLC2A10 variant and CNO, more rigorous animal model is required.

GLUT10 mainly functions as an ascorbate transporter in intracellular organelles such as mitochondria (Biochem Biophys Res Commun, 2026 815:153676; Hum Mol Genet, 2010, 19:3721). This is not congruent to the contention that the observed cellular changes are due to increased glucose transport function of the SLC2A10 variant in the submitted paper. Therefore, whether GLUT10 transporter is present in the plasma membrane of macrophages and osteoclasts, and whether its role as an ascorbate transporter is not involved should be clarified.

The mechanism for the glycolysis regulation by SLC2A10 variant protein (D292E) is not clear. Expression of D292E affected transcriptome, which showed changes in genes of the glycolytic pathway. What would be a link between plasma membrane GLUT10 (if GLUT10 works as a glucose transporter as suggested in this manuscript) to gene transcription?

Previous report (Hum Mol Genet, 2010, 19:3721) showed that loss-of-function of SLC2A10 increased ROS. This is contradictory to the increased ROS by gain-of-function variation of SLC2A10 in the submitted paper. How could this contradiction be resolved?

In the zebrafish experiment, neutral red staining is not a direct and conclusive method to assess macrophages. In addition, Alizarin Red staining does not show convincing differences between samples. In the text, Fig. 7D is stated to show cell counting data, which is not present in the figure.

Referee #1 Review

Remarks for Author:

In this manuscript, the authors describe the detection of a variant in SLC2A10 as a novel disease-relevant variant in chronic non-bacterial osteomyelitis and provide quite a few experimental data of models to show the molecular effect. They describe molecular impacts of the candidate variant, while their interpretation is partly overdone, expressed as definitive and their discussion rather unstructured. Major aspects are the following ones:

- CNO is not only an inflammatory disease of the bone and skin, but also intestine and lung. These clinical aspects are important to assess. Did any of the family members have symptoms of these organs?
- The clinical details of the patients, especially the older ones is rather scarce and limited. When did the father/ the grandmother develop acne? Did they have similar CNO at the mandibular region in childhood? At what age did they develop CNO? How about the previous generations, any anamnestic details?
- The pedigree in figure 1 provides evidence of a further affected individuals in generation I. Why was this patient not genetically tested? The information on wildtype/ variant carriers provided is not correct. COMPLETE segregation is not performed here, cosegregation is not the correct wording. Probably in figure S2, variants is meant not genes.
- There is no accumulation of nuclei in the bone scan, but radionuclide.
- Introduction: GLUT1 is another protein than GLUT10 and should not be mixed up, probably just omitted.
- Considering the number of genomic variants that are known in general, it is more appropriate to use different wording: instead of "mutation" "variant" should be used throughout the manuscript. Gene names need italic print, protein names do not.
- Which classification criteria did the authors use to classify variants in genes partly unrelated to any phenotype/ disease as pathogenic/ likely pathogenic? Being a heterozygous carrier of a variant that is disease-causing in case of a 2nd variant on the other allele is a common finding, but not related to disease. What was the molecular basis for that? Although the variants are rare and might have protein predictions suggesting pathogenicity, that does not mean that they are (likely) pathogenic. These aspects need to be discussed in that sense.
- The reviewer does not know the wording of "morphogenetic pathway". What is meant here? Using only "pathway" is probably enough. Also, the reviewer is not aware of literature that describes that metabolic pathways, pyrimidine metabolism, sphingolipid metabolism, and amino acid biosynthesis are central in CNO.
- It is not shown in the manuscript that the hydrogen bonds are critical, therefore this wording needs to be less interpretative.
- The authors use a protein array assessing 40 different cytokines of an inflammation array. Those represent players in certain inflammatory pathways. Therefore, an enrichment analysis is not appropriate.
- The nomenclature of variants is not uniform and standardized. The authors should use the international standard of HGVS nomenclature. Also, it should be differentiated whether the nucleotide level or the protein level is meant; e.g. when a vector is mutated, the nucleotide level needs to be given.
- The abbreviation "NC" is used many times, not explained, but means different things. The authors

need to explain in single experimental settings what is meant by it.

- In the results on p. 6: the authors argue that an increase in protein amount indicates that the variant is acquired. The reviewer cannot follow this line of causality. Also, this increase is only marginal.
- Last sentence 1st paragraph p. 6 is not a result and should be omitted.
- How was the immunofluorescence quantified? How many cells/ slides were assessed?
- How were the osteoclasts quantified? Regarding the multi nuclei, this is not trivial. How many cells/ slides were assessed?
- The differences in macrophage infiltration of the mandibular region shown in Fig. 7C is not recognizable by the picture provided. Was the resolution different in microscopy? Small graphical items might underline and demonstrate what is meant here.
- Where all experiments shown in figure 6 performed under stimulation with LPS? What means abbreviation ECAR? Not explained at all.
- Transcriptomic analyses p. 7: what type of tissue was the source of the RNA used for these analyses?
- The authors did not use a threshold for logfold changes in the transcriptomic analyses, although this might result in rather subtle changes that are biologically not relevant.
- P. 8: the authors point out to pathways that are only represented by very few genes, but do not mention other ones or point to ones with rather high p-values. That is rather doubtful.
- Probably, the PCR products were not recombined but amplified in *E. coli*, p. 21.
- Figure legends are mainly rather short and scarce and do not provide enough details to understand them. Figure 1: is the residue of variant shown in green? For the review process, small icons indicating the figure no. are very helpful.
- The discussion of the manuscript is not in order, partly not logically structured and confuse. It needs thorough re-organization and rephrasing to be easier to understand. The results and discussion section would benefit by expressing the findings in a more subtle, cautious manner. Not all results given are so definite.
- Considering the role of SLC2A10 in arterial tortuosity, did the authors find any evidence that their patients have symptoms of the described vascular spectrum? How can it be explained that some variants in the gene cause a skeletal/ dermatological phenotype, and other a vascular one, or in other words is there evidence for a genotype phenotype correlation?

Minor aspects that need consideration:

- To include page numbers in a manuscript is really helpful to discuss potential changes.
- There are no "hereditary families". Also the use of "hydrogen bonding" should be replaced by hydrogen bonds, or Western blotting by Western blot, mutation primer sequences by sequences of primers used for mutagenesis. Wording conservatism? In general, the use of a professional editorial service would substantially improve the readability of the manuscript.
- Figure 8A on p. 10, 3rd paragraph: probably Fig. 7A is meant.
- Fig. part 6D is not mentioned in the text.
- Supplement: Figure/ table titles should be on the same page as the figure/ table itself.
- What is meant by GRCh3 on p. 17?
- Functional genomic analyses p. 16: probably, functional analyses are meant

Referee #2 Review

Comments on Novelty/Model System for Author:

The medical impact is limited by the one family investigated. Authors should consider screening larger cohorts for gene variants which may increase the impact of their findings dramatically. Furthermore, the authors should try to explain differential phenotypes associated with SLC2A10 variants, namely CNO versus arterial tortuosity syndrome. Are different parts functional domains of the protein affected?

Remarks for Author:

The authors report an interesting family with a cluster of CNO patients linked to SLC2A10 GOF variants. The molecular investigations are compelling and add to the literature, but several points should be addressed:

- Grammar and style could benefit from additional attention. In some sections of the results, authors discuss findings without putting discussion in the context of the literature. What is 'cutaneous warmth'? I am certain the authors mean to say 'local warmth and swelling over affected bones'?
- The story around SLC2A10 GOF is interesting and compelling, I strongly suggest deleting the very superficial information from other families. This could/should be an independent story including clinical and functional data.
- Mandibular involvement in CNO is NOT 'common'; it is indeed rare, e.g. PMID: 38279375
- What treatment could be beneficial, what were patients treated with, would in vitro inhibition be an appropriate tool to predict targeted treatments. This family and developed tools would be wonderful candidates to test 'new' interventions. This would add significantly to the story presented.
- The authors are missing key references when discussing the role of macrophages and their inflammatory phenotype (PMID: 38401466; PMID: 33314777) as well as osteoclasts (PMID: 37315560) in CNO. All these are linked to genetic variants that have not been discussed by the authors.
- This referee suggests altering the order of results. It may be beneficial to discuss macrophage function first and the move to all osteoclast data together.
- Coming back to gene variants in CNO versus arterial tortuosity syndrome. Would it be possible to test a variant identified in arterial tortuosity as well to understand potential differences on inflammatory signaling and osteoclast function?
- As mandibular CNO is rare and the here reported SLC2A10 GOF variant is 'new', would it be possible to test this gene in a cohort of CNO patients to see whether it plays a role in more CNO patients with versus without mandibular involvement? Are there further associations, e.g. with high inflammatory markers, etc? What would be the benefit of knowing? Again, this links to treatment response and possible targeted treatments.

Referee #3 Review

Comments on Novelty/Model System for Author:

Human sample staining lacks the description of sample collection methods and relevant ethical approvals

Remarks for Author:

This study reports a novel gain-of-function mutation in the SLC2A10 gene associated with chronic nonbacterial osteomyelitis and explores the mechanism by which it drives macrophage polarization and osteoclast differentiation through glucose metabolism reprogramming. The research offers a new insight into the pathogenesis of CNO combining clinical genetics, cell models, zebrafish models, and multi-omics analysis. However, despite the novelty and potential importance of this study, several issues remain to be addressed. The main problem lies in the insufficient depth of presentation and interpretation of some key data, which affects the rigor and persuasiveness of the conclusions. Suggested revisions are as follows.

1. The main conclusions are based on a small family with only three patients. Such a small sample size makes it difficult to completely rule out the possibility that this rare variant is familial idiopathic or a chance co-segregation with the disease. The authors identified other candidate genes in two other families, but did not explore the presence of SLC2A10 mutations in these families or rule out their role, which weakens the assertion that SLC2A10 is a major pathogenic gene.
2. Although this study characterized the mutation as "gain of function," it lacks experimental data in cell or animal models demonstrating that knocking down or eliminating endogenous SLC2A10 does not produce a phenotype similar to that of mutant overexpression to support the "gain of function" assessment. Further experiments are needed.
3. It is noted that the manuscript contains data on human sample staining, this study needs to supplement the description of sample collection methods and relevant ethical approvals.

4. The biological models used in this study were inconsistent, including human samples, mouse-derived RAW264.7 cells, and zebrafish animal models. Although this gene mutation is highly conserved, the use of multiple species models reduces the persuasiveness of the experiment. Please explain this.

5. It is suggested that the D292E mutation alters the glucose transport activity of GLUT10. Structural simulations suggest this possibility, but direct experimental evidence, such as direct glucose uptake assays to compare the transport rate of WT and MT GLUT10 in macrophages is lacking.

6. This study has observed the simultaneous occurrence of enhanced glucose metabolism and increased inflammation/osteoclast differentiation, but have failed to demonstrate that the former is the cause of the latter. The key experiment is to determine whether the pro-inflammatory and osteoclast-promoting effects induced by mutant SLC2A10 are reversed after the use of glycolysis inhibitors. Under the current circumstances, the "glycolysis-driven" conclusion appears to be more of a correlation than a causal relationship.

7. In zebrafish, it is recommended to supplement with experiments, such as knocking down endogenous *slc2a10* in zebrafish, or constructing a point mutation knock-in model using CRISPR/Cas9 to avoid the off-target effects brought by mRNA overexpression.

8. This study only described the phenotypes of macrophage polarization, osteoclast differentiation, and the effects on osteoblasts, without explaining the mechanisms involved. For example, the mechanisms of inflammatory osteoclast differentiation and the intermediate mediators in osteoclast/osteoblast co-culture were not further explored, which reduced the depth of the study.

9. The grammar and accuracy of the manuscript need to be thoroughly corrected, such as the standardization of gene naming.

Dear Editor:

Thank you for considering our manuscript entitled "A Novel SLC2A10 Gain-of-Function Mutation Links Glycolytic Macrophage Polarization to Chronic Nonbacterial Osteomyelitis" for possible publication in *EMBO MOLECULAR MEDICINE*.

We are grateful for the reviewers' comments that helped have improved our manuscript.

All issues raised by them were addressed and the recommended changes made are highlighted in RED in the revised manuscript.

The aforementioned are detailed below together with responses to their concerns.

Reviewer 1#

In this manuscript, the authors describe the detection of a variant in *SLC2A10* as a novel disease-relevant variant in chronic non-bacterial osteomyelitis and provide quite a few experimental data of models to show the molecular effect. They describe molecular impacts of the candidate variant, while their interpretation is partly overdone, expressed as definitive and their discussion rather unstructured. Major aspects are the following ones.

Response: Thank you for your encouragement and invaluable suggestions. We will endeavor to address the concerns raised to the best of our abilities.

1. CNO is not only an inflammatory disease of the bone and skin, but also intestine and lung. These clinical aspects are important to assess. Did any of the family members have symptoms of these organs?

Response: Thank you for this important comment. We agree that CNO may involve extra-osseous organs such as the intestine and lung. We carefully reviewed the clinical records and medical history of all affected family members. None of the patients exhibited gastrointestinal or respiratory symptoms, and no evidence of intestinal or pulmonary involvement was identified during follow-up. We have clarified this in the revised manuscript.

2. The clinical details of the patients, especially the older ones is rather scarce and limited. When did the father/ the grandmother develop acne? Did they have similar CNO at the mandibular region in childhood? At what age did they develop CNO? How about the previous generations, any anamnestic details?

Response: Thank you for this insightful comment. We agree that detailed longitudinal clinical information would be highly valuable. We made substantial efforts to retrospectively obtain additional data from the older family members; however, due to the long time interval, geographic distance, and age-related recall limitations, precise historical details and further clinical evaluations were not feasible.

The father reported experiencing lumbar pain during his youth, which may suggest possible skeletal involvement, but no imaging or medical documentation from that period is available. The grandmother reportedly had severe acne in her youth accompanied by tenderness at certain skeletal sites. However, the exact distribution, timing, and diagnostic confirmation of these symptoms could not be reliably established.

Given these constraints, more detailed anamnestic information and objective verification of past manifestations could not be obtained.

3. The pedigree in figure 1 provides evidence of a further affected individuals in generation I. Why was this patient not genetically tested? The information on wildtype/ variant carriers provided is not correct. COMPLETE segregation is not performed here, cosegregation is not the correct wording. Probably in figure S2, variants is meant not genes.

Response: Thank you for this important comment. We fully agree with the reviewer that sequencing of generation I would strengthen the evidence and make our study more convincing. However, the affected individual in generation I was not genetically tested because she lives far from our center and, due to advanced age and limited mobility, it was not feasible to obtain a DNA sample. Therefore, genetic analysis could not be performed in this individual.

We acknowledge that complete segregation analysis across all generations was not achievable in this family. Genetic testing was performed in the available affected and unaffected members, and the identified variant was consistently present in affected individuals and absent in unaffected tested relatives. While this supports intrafamilial segregation, we agree that this does not constitute complete segregation across the entire pedigree.

We will revise the manuscript to correct the description of wild-type and variant carriers and to avoid overstatement regarding segregation analysis (Page4,15-16) . In addition, we will correct “genes” to “variants” in Figure S2.

4. There is no accumulation of nuclei in the bone scan, but radionuclide.

Response: We sincerely thank the reviewer for pointing out this issue. We have revised it in the manuscript (Fig. 1F).

5. Introduction: GLUT1 is another protein than GLUT10 and should not be mixed up, probably just omitted.

Response: We thank the reviewer for this helpful suggestion. To avoid any possible misunderstanding, the discussion of GLUT1 has been removed from the manuscript.

6. Considering the number of genomic variants that are known in general, it is more appropriate to use different wording: instead of "mutation" "variant" should be used throughout the manuscript. Gene names need italic print, protein names do not.

Response: Thank you for your helpful suggestion. We have replaced “mutation” with “variant” throughout the manuscript and corrected the formatting of gene and protein names accordingly.

7. Which classification criteria did the authors use to classify variants in genes partly unrelated to any phenotype/ disease as pathogenic/ likely pathogenic? Being a heterozygous carrier of a variant that is disease-causing in case of a 2nd variant on the other allele is a common finding, but not related to disease. What was the molecular basis for that? Although the variants are rare and might have protein predictions suggesting pathogenicity, that does not mean that they are (likely) pathogenic. These aspects need to be discussed in that sense.

Response: We thank the reviewer for this important comment. Variant classification was performed according to the ACMG guidelines. We have revised the Methods section to clearly describe the classification criteria and the autosomal dominant inheritance model used in our analysis. We also clarified that rarity and in silico predictions were used as supporting evidence rather than sole determinants of pathogenicity.

8. The reviewer does not know the wording of "morphogenetic pathway". What is meant here? Using only "pathway" is probably enough. Also, the reviewer is not aware of literature that describes that metabolic pathways, pyrimidine metabolism, sphingolipid metabolism, and amino acid biosynthesis are central in CNO.

Response: Thank you for this helpful suggestion. We appreciate the reviewer’s comment regarding the wording of “morphogenetic pathway” and the relevance of the mentioned metabolic pathways to CNO.

Following this comment and a related suggestion from another reviewer, we have streamlined the manuscript and removed the additional data from the other two families, including the corresponding pathway analysis. This was done to avoid potential confusion and to maintain focus on the main findings and central narrative of the study.

9. It is not shown in the manuscript that the hydrogen bonds are critical, therefore this wording needs to be less interpretative.

Response: Thank you for this insightful comment. We agree that our original wording was overly interpretative, as the current data do not directly demonstrate that these hydrogen bonds are functionally critical. We have therefore revised the text to use more cautious language and to avoid implying a definitive functional role. The revised sentence now reads: “Structural modeling suggested that in the wild-type GLUT10 protein, D292 is predicted to form hydrogen bonds with adjacent transmembrane residues. The D292E substitution is predicted to alter two hydrogen bond interactions, which may influence local structural stability.” We believe this modification more accurately reflects the limitations of structural modeling predictions.

10. The authors use a protein array assessing 40 different cytokines of an inflammation array. Those represent players in certain inflammatory pathways. Therefore, an enrichment analysis is not appropriate.

Response: We thank the reviewer for this important and constructive comment. We agree that the cytokine array represents a predefined inflammation-focused panel rather than an unbiased proteomic screen, and therefore formal pathway enrichment analysis is not statistically appropriate.

In response to this concern, we have removed the KEGG and GO enrichment analyses from the manuscript and revised the Results section accordingly. The cytokine data are now presented descriptively using heatmap visualization, without implying statistical pathway enrichment.

We believe this revision improves the methodological rigor and clarity of the study.

11. The nomenclature of variants is not uniform and standardized. The authors should use the international standard of HGVS nomenclature. Also, it should be differentiated whether the nucleotide level or the protein level is meant; e.g. when a vector is mutated, the nucleotide level needs to be given.

Response: We thank the reviewer for this important and constructive comment. We agree that variant nomenclature should be uniform and follow the international Human Genome Variation Society (HGVS) standards.

In response, we have carefully revised the entire manuscript to ensure consistent and standardized variant descriptions throughout. All variants are now reported according to HGVS recommendations, with clear distinction between nucleotide-level changes (denoted by “c.”) and protein-level changes (denoted by “p.”).

Specifically, the variant is presented at its first occurrence as c.876C>A (p.Asp292Glu) based on NM_030777.4, and the protein change is subsequently abbreviated as p.D292E throughout the manuscript for consistency.

In addition, when describing plasmid construction, lentiviral generation, and in vitro transcription experiments, the nucleotide-level change (c.) is explicitly provided, as recommended.

We believe these revisions improve the clarity, accuracy, and compliance of the manuscript with HGVS standards.

12. The abbreviation "NC" is used many times, not explained, but means different things. The authors need to explain in single experimental settings what is meant by it.

Response: We thank the reviewer for this helpful comment. We agree that the abbreviation “NC” was used inconsistently and may have caused confusion across different experimental settings.

In response, we have replaced “NC” with precise and context-specific terminology throughout the manuscript. In the RAW264.7 experiments, the lentiviral empty vector group is now consistently referred to as the “EV” group. In the zebrafish experiments, control embryos are now clearly designated as “Control”, and this has been explicitly defined in each figure legend and corresponding Methods section.

These revisions ensure clarity and consistency in the description of experimental controls.

13. In the results on p. 6: the authors argue that an increase in protein amount indicates that the variant is acquired. The reviewer cannot follow this line of causality. Also, this increase is only marginal.

Response: We thank the reviewer for this important comment. We agree that increased protein abundance alone cannot support the conclusion that a variant is acquired, and we have therefore removed the previous statement suggesting this causal relationship.

To further clarify the functional impact of the c.876C>A (p.Asp292Glu) variant, we performed glucose uptake assays in four experimental groups (Empty Vector, KO, KO+WT, and KO+D292E). As expected, *SLC2A10* knockout reduced glucose uptake, and re-expression of wild-type *SLC2A10* partially restored glucose uptake. Importantly, the KO+D292E group exhibited significantly higher glucose uptake compared with the KO+WT group (EV3C), indicating enhanced functional activity of the variant.

2-DG uptake in RAW cells

EV3 C: Glucose uptake assay demonstrating a significantly higher glucose uptake rate in D292E macrophages compared with WT and empty vector (EV) controls. p-values: * $p < 0.05$, ** $p < 0.01$, *** $p < 0.001$; ns: no significant difference.

Western blot analysis showed that protein expression in the KO+D292E group was slightly higher than in the KO+WT group. However, the magnitude of the increase in glucose uptake exceeded the modest difference in protein abundance, suggesting that the enhanced glucose uptake cannot be explained solely by increased protein levels.

Based on these functional data, we now describe the p.Asp292Glu variant as exhibiting a gain-of-function-like effect rather than referring to it as an acquired mutation. The manuscript has been revised accordingly.

14. Last sentence 1st paragraph p. 6 is not a result and should be omitted.

Response: We thank the reviewer for this helpful comment. We agree that the sentence in question reflects a general interpretation rather than a direct experimental result. In response, we have removed this statement from the Results section to maintain a clear distinction between objective findings and interpretative discussion.

15. How was the immunofluorescence quantified? How many cells/ slides were assessed?

Response: We thank the reviewer for this important comment. We have now clarified the immunofluorescence quantification procedure in the revised manuscript.

Briefly, ROS fluorescence was assessed in RAW264.7 cells cultured in 6-well plates. For each experimental group, three independent wells were used as biological replicates. To minimize sampling bias, each well was divided into predefined regions, and images were captured systematically under a 40 \times objective from each region using identical exposure settings across all groups.

Fluorescence intensity was quantified using ImageJ software (NIH, USA). After background subtraction, the mean fluorescence intensity (MFI) was measured for each image. Cell density was maintained as consistently as possible across wells to ensure comparability. The average fluorescence intensity per well was calculated and used for statistical analysis.

These methodological details have been added to the Methods section to improve clarity and reproducibility.

16. How were the osteoclasts quantified? Regarding the multi nuclei, this is not trivial. How many cells/ slides were assessed?

Response: We thank the reviewer for this important comment. The quantification procedures for TRAP staining in both human mandibular tissue samples and RAW264.7 cells have now been clarified in the revised manuscript.

For human tissue sections, at least three non-consecutive sections per specimen were analyzed. From each section, five randomly selected high-power fields were imaged under identical exposure settings. TRAP-positive multinucleated cells (defined as cells containing two or more nuclei with positive cytoplasmic staining) were manually counted. The mean number of TRAP-positive cells per field was first calculated for each section and then averaged to obtain a single value per specimen for statistical analysis.

For in vitro osteoclast differentiation assays, RAW264.7 cells were cultured in 6-well plates, with three independent wells per experimental group serving as biological replicates. Each well was divided into nine predefined regions to ensure systematic sampling, and images were captured from each region under identical magnification settings. TRAP-positive multinucleated cells (≥ 2 nuclei) were manually counted in all imaged regions, and the total number per well was used for statistical analysis.

These methodological details have been incorporated into the Methods section to ensure transparency and reproducibility.

17. The differences in macrophage infiltration of the mandibular region shown in Fig. 7C is not recognizable by the picture provided. Was the resolution different in microscopy? Small graphical items might underline and demonstrate what is meant here.

Response: Thank you for your valuable suggestion. The red dots represent macrophages. To improve clarity, we have added arrows in the original image to specifically indicate the macrophages.

Figure7 (C) Neutral red staining marks macrophages in two groups of mandibles in 12 dpf zebrafish. (The white dashed line delineates the jawbone region. Macrophages are shown in red and are indicated by yellow arrows.)

18. Where all experiments shown in figure 6 performed under stimulation with LPS? What means abbreviation ECAR? Not explained at all.

Response: Thank you for your careful review and valuable comments.

Regarding Figure 6, panels 6A and 6C were performed under LPS stimulation, whereas Figure 6B was conducted under RANKL stimulation. We agree that this was not sufficiently clear in the original figure legend. To avoid confusion, we have revised the figure legend to explicitly state the stimulation conditions for each panel.

Concerning the abbreviation ECAR (Extracellular Acidification Rate), it has been defined in the main text. However, we acknowledge that it was not explained in the figure legend, which may cause inconvenience for readers. We have now added the full term and its explanation in the revised figure legend for clarity.

We appreciate the reviewer's insightful suggestions, which have helped us improve the clarity of the manuscript.

19. Transcriptomic analyses p. 7: what type of tissue was the source of the RNA used for these analyses?

Response: Thank you for this important question.

The RNA used for the transcriptomic analyses was extracted from whole-body homogenates of 4-day-post-fertilization (4 dpf) zebrafish larvae. For each experimental group, 15 larvae were pooled together for total RNA extraction to ensure sufficient RNA yield and to minimize biological variability.

Thank you for helping us improve the methodological transparency of our manuscript.

20. The authors did not use a threshold for logfold changes in the transcriptomic analyses, although this might result in rather subtle changes that are biologically not relevant.

Response: We thank the reviewer for this insightful comment.

First, we would like to clarify that in Figure 8C the label “p value” was a typographical error. The analysis was in fact performed using adjusted p values (q values) throughout. All enrichment analyses were conducted based on multiple-testing corrected p values (FDR-adjusted), which provides robust statistical control and minimizes false-positive findings.

Regarding the absence of a log₂ fold-change threshold, our primary objective was to capture coordinated transcriptional shifts at the pathway level rather than focusing solely on genes with large fold changes. It is well established that biologically meaningful pathway alterations may arise from modest but consistent changes across multiple genes. Therefore, we prioritized statistical significance (adjusted p value) and pathway-level enrichment over imposing an arbitrary fold-change cutoff.

21. P. 8: the authors point out to pathways that are only represented by very few genes, but do not mention other ones or point to ones with rather high p-values. That is rather doubtful.

Response: We thank the reviewer for this careful observation.

We acknowledge that in the previous version of the manuscript, certain pathways highlighted in the GSEA results were represented by a relatively small number of genes, and some pathways with comparatively higher p-values were not sufficiently discussed. We understand that this presentation may have raised concerns regarding selective reporting.

After careful reconsideration, we have removed the GSEA figure and the related interpretation from the revised manuscript to avoid potential overinterpretation of results that may not be sufficiently robust. We agree that pathways represented by only a few genes or with marginal statistical significance should be interpreted with caution.

In the revised version, we focus on analyses with stronger statistical support and clearer biological relevance. This modification improves the rigor and transparency of the study.

We appreciate the reviewer’s comment, which helped us refine the presentation of our results.

22. Probably, the PCR products were not recombined but amplified in *E. coli*, p. 21.

Response: We thank the reviewer for this comment.

We apologize for the lack of clarity in the description of the cloning procedure. After subcloning the coding sequences of wild-type and D296E mutant *slc2a10* into the pcDNA3.1 vector, the recombinant plasmids were transformed into *E. coli* for amplification. Plasmids were then purified and verified by sequencing before being used as templates for in vitro transcription.

We have revised the Methods section (Page 19) to clarify this point.

We appreciate the reviewer’s careful reading, which helped us improve the accuracy of the methodological description.

23. Figure legends are mainly rather short and scarce and do not provide enough details to understand them. Figure 1: is the residue of variant shown in green? For the review process, small icons indicating the figure no. are very helpful.

Response: We thank the reviewer for this helpful suggestion.

We have revised and expanded all figure legends to provide more detailed descriptions, including sufficient methodological and interpretative information to ensure that the figures can be understood independently of the main text. For Figure 2 specifically, we have clarified that the variant residue is highlighted in green in the structural model.

In addition, small figure number indicators have been added where appropriate to improve clarity during the review process.

We appreciate the reviewer's constructive comments, which have helped improve the clarity and presentation of the manuscript.

24. The discussion of the manuscript is not in order, partly not logically structured and confuse. It needs thorough re-organization and rephrasing to be easier to understand. The results and discussion section would benefit by expressing the findings in a more subtle, cautious manner. Not all results given are so definite.

Response: We sincerely thank the reviewer for this constructive and insightful comment. We agree that the original Discussion section lacked optimal logical flow and clarity in certain parts.

Accordingly, we have thoroughly reorganized and restructured the Discussion section to improve its coherence and readability. Specifically, we:

- 1) Reordered the paragraphs to better align with the sequence of the key findings.
- 2) Clearly separated the interpretation of results from methodological considerations.
- 3) Improved transitions between sections to enhance logical continuity.
- 4) Removed redundant statements and clarified ambiguous expressions.

In addition, we carefully revised the wording throughout the Results and Discussion sections to adopt a more cautious and balanced tone. Statements that were previously overly definitive have been moderated to better reflect the exploratory nature and limitations of the study.

We believe these revisions have substantially improved the clarity, logical structure, and scientific rigor of the manuscript.

25. Considering the role of SLC2A10 in arterial tortuosity, did the authors find any evidence that their patients have symptoms of the described vascular spectrum? How can it be explained that some variants in the gene cause a skeletal/ dermatological phenotype, and other a vascular one, or in other words is there evidence for a genotype phenotype correlation?

Response: We thank the reviewer for raising this important question.

In our cohort of CNO patients carrying the *SLC2A10* variant, we did not observe any clinical features consistent with the vascular spectrum typically described in arterial tortuosity syndrome (ATS), such as arterial tortuosity, aneurysm formation, or other major vascular abnormalities. Detailed clinical evaluation and imaging examinations did not reveal evidence of vascular involvement in these patients.

Regarding genotype–phenotype correlation, previously reported SLC2A10 variants associated with arterial tortuosity syndrome are predominantly loss-of-function mutations, including nonsense, frameshift, or splice-site variants that are predicted to severely impair or abolish GLUT10 function. In contrast, the variant identified in our study is a missense mutation affecting a different residue from those classically reported in ATS. This suggests that the molecular consequence of the variant may be distinct from the loss-of-function mechanisms underlying the vascular phenotype.

Taken together, available evidence supports the possibility of genotype–phenotype correlation in SLC2A10-related disorders. Severe loss-of-function variants appear to be associated with the vascular phenotype observed in ATS, whereas specific missense variants may result in altered or partial functional changes that manifest predominantly as skeletal and/or dermatological features without overt vascular involvement.

Minor aspects that need consideration:

26. To include page numbers in a manuscript is really helpful to discuss potential changes.

Response: We fully agree that including page numbers facilitates the review process. Page numbers have now been added throughout the revised manuscript.

27. There are no "hereditary families". Also the use of "hydrogen bonding" should be replaced by hydrogen bonds, or Western blotting by Western blot, mutation primer sequences by sequences of primers used for mutagenesis. Wording conservatism? In general, the use of a professional editorial service would substantially improve the readability of the manuscript.

Response: We appreciate the reviewer's attention to precise terminology. The suggested corrections have been implemented, including replacing "hereditary families" with appropriate wording (TableS1 title), "hydrogen bonding" with "hydrogen bonds," "Western blotting" with "Western blot," and "mutation primer sequences" with "Primers used for cloning and site-directed mutagenesis."

In addition, the manuscript has undergone careful language revision for consistency and clarity. We have also sought professional editorial assistance to further improve readability.

28. Figure 8A on p. 10, 3rd paragraph: probably Fig. 7A is meant.

Response: We thank the reviewer for noting this inconsistency. The reference has been corrected to Figure 7A.

29. Fig. part 6D is not mentioned in the text.

Response: The corresponding description of Figure 6D has now been added to the main text to ensure completeness.

30. Supplement: Figure/ table titles should be on the same page as the figure/ table itself.

Response: As suggested, all supplementary figure and table titles have been adjusted to appear on the same page as their respective figure or table.

31. What is meant be GRCh3 on p. 17?

Response: We apologize for this typographical error. It has been corrected to "GRCh38."

32. Functional genomic analyses p. 16: probably, functional analyses are meant

Response: We thank the reviewer for pointing this out. The term has been corrected to "functional analyses."

Reviewer 2#

The medical impact is limited by the one family investigated. Authors should consider screening larger cohorts for gene variants which may increase the impact of their findings dramatically.

Furthermore, the authors should try to explain differential phenotypes associated with *SLC2A10* variants, namely CNO versus arterial tortuosity syndrome. Are different parts functional domains of the protein affected?

Response: We sincerely thank the reviewer for these valuable comments. We fully agree that expanding the cohort size and screening additional patients for *SLC2A10* variants would substantially strengthen the generalizability and clinical impact of our findings. We acknowledge that our study is based on a single family, which limits generalizability. However, CNO is a rare and genetically heterogeneous disorder, and our findings are supported by functional validation. We agree that screening larger independent cohorts would strengthen the clinical relevance, and this has now been stated as a limitation and future direction in the revised Discussion.

Most previously reported *SLC2A10* variants associated with arterial tortuosity syndrome (ATS) are loss-of-function mutations (e.g., nonsense or frameshift), leading to severe impairment of GLUT10. In contrast, the variant identified in our CNO family is a missense mutation affecting a different residue and is unlikely to cause complete loss of function.

This suggests a possible genotype–phenotype correlation, where severe loss-of-function variants result in vascular manifestations, whereas specific missense variants may cause partial functional alteration and a predominantly skeletal/inflammatory phenotype.

We appreciate the reviewer's insightful suggestions, which have helped improve the manuscript. We have now incorporated the relevant discussion into the Discussion section of the revised manuscript.

1. Grammar and style could benefit from additional attention. In some sections of the results, authors discuss findings without putting discussion in the context of the literature. What is 'cutaneous warmth'? I am certain the authors mean to say 'local warmth and swelling over affected bones'?

Response: We sincerely appreciate the reviewer's careful and constructive comments.

Regarding grammar and style, the manuscript has been carefully revised and polished throughout to improve language accuracy, consistency, and readability. We have paid close attention to grammatical structures and academic writing style.

For the description in the Results section, we have revised the text to better integrate and contextualize our findings with previous literature, ensuring that the results are appropriately discussed in the context of existing knowledge.

Concerning the term "cutaneous warmth", we agree that this description was ambiguous. We have revised it to "local warmth and swelling over affected bones" as suggested by the reviewer (page2), to accurately reflect the clinical phenotype.

We greatly appreciate these valuable suggestions, which have significantly improved the quality and clarity of the manuscript.

2. The story around *SLC2A10* GOF is interesting and compelling, I strongly suggest deleting the very superficial information from other families. This could/should be an independent story including clinical and functional data.

Response: We sincerely thank the reviewer for the positive comments regarding the *SLC2A10* GOF findings and for the constructive suggestion.

We agree that the GOF-related observations may represent an independent and more comprehensive study integrating detailed clinical and functional data. In line with the reviewer's recommendation, we have removed the superficial information from the other families in order to improve the focus, clarity, and coherence of the current manuscript.

We believe that this revision strengthens the overall presentation of the study and better highlights the central findings.

We appreciate the reviewer's thoughtful advice, which has helped us substantially improve the manuscript.

3. Mandibular involvement in CNO is NOT 'common'; it is indeed rare, e.g. PMID: 38279375

Response: We sincerely thank the reviewer for pointing this out and for providing the relevant reference (PMID: 38279375).

We agree that mandibular involvement in CNO is not common and is considered relatively rare. The statement has been corrected accordingly in the revised manuscript, and the suggested reference has been added to support this point.

We appreciate the reviewer's careful attention to accuracy.

4. What treatment could be beneficial, what were patients treated with, would in vitro inhibition be an appropriate tool to predict targeted treatments. This family and developed tools would be wonderful candidates to test 'new' interventions. This would add significantly to the story presented.

Response: We sincerely thank the reviewer for this insightful and translationally oriented suggestion.

At present, the standard treatment for CNO includes nonsteroidal anti-inflammatory drugs (NSAIDs) as first-line therapy, followed by corticosteroids, immunosuppressants, bisphosphonates, or TNF- α inhibitors in refractory cases. The affected individuals in our study were managed according to current clinical practice guidelines. Since 2018, our

center has treated several hundred patients with CNO in routine clinical practice. Intravenous zoledronic acid has been commonly administered as part of our treatment strategy, including in the affected members of this family.

Importantly, our study aims to move beyond symptomatic treatment by identifying disease-driving mechanisms that may serve as therapeutic entry points. Our functional analyses suggest that the *SLC2A10* variant induces metabolic reprogramming characterized by enhanced glycolytic activity. Notably, pharmacological inhibition of glycolysis using 2-deoxy-D-glucose (2-DG) partially rescued the abnormal cellular phenotype in vitro. These findings support the concept that metabolic modulation could represent a rational targeted strategy in this genetic context.

EV3C: M1 polarization markers were analyzed in the following groups (Vector: Empty Vector cells, WT: Overexpression of wild-type *SLC2A10* cells, D292E: Overexpression of variant *SLC2A10* cells; KO: *SLC2A10* knockout cells; WT-KO: *SLC2A10* knockout cells expressing wild-type *SLC2A10*; D292E-KO: *SLC2A10* knockout cells expressing the *SLC2A10* D292E variant)

However, while the in vitro rescue experiments provide proof-of-principle evidence, we believe that direct clinical translation at this stage would be premature. To ensure safety and biological validity, we are currently extending these findings into in vivo models to evaluate therapeutic efficacy and potential adverse effects.

We agree with the reviewer that this family and the established experimental platform offer a valuable opportunity to test novel targeted interventions in a genetically defined setting. We have expanded the Discussion to highlight this translational perspective and future directions.

We greatly appreciate this comment, which strengthens the clinical and therapeutic implications of our work.

5. The authors are missing key references when discussing the role of macrophages and their inflammatory phenotype (PMID: 38401466; PMID: 33314777) as well as osteoclasts (PMID: 37315560) in CNO. All these are linked to genetic variants that have not been discussed by the authors.

Response: We sincerely thank the reviewer for highlighting these important references.

We agree that macrophage inflammatory phenotypes and osteoclast dysregulation play a critical role in the pathogenesis of CNO, and that recent studies (PMID: 38401466; PMID: 33314777; PMID: 37315560) have linked these mechanisms to specific genetic variants. We regret that these key publications were not sufficiently discussed in the original version of the manuscript.

In the revised manuscript, we have incorporated these references and expanded the Discussion to better contextualize our findings within the broader framework of genetically driven innate immune dysregulation and osteoclast-mediated bone inflammation in CNO. We now discuss how our observations may intersect with previously reported variant-associated macrophage activation and osteoclast functional alterations.

We appreciate the reviewer's careful reading and valuable suggestion, which has significantly strengthened the scientific context of our study.

6. This referee suggests altering the order of results. It may be beneficial to discuss macrophage function first and the move to all osteoclast data together.

Response: We sincerely thank the reviewer for this thoughtful suggestion regarding the organization of the Results section. We apologize for the misleading wording in the previous version. The legend of Figure 6 has been corrected accordingly.

In the current structure, Figures 4–6 are arranged to follow a progressive framework from phenotype to mechanism. Specifically, we first present the inflammatory phenotype and macrophage functional alterations (Figure 4), followed by osteoclast-related findings (Figure 5), and finally the mechanistic investigations focusing on macrophage polarization and osteoclast differentiation (Figure 6). We chose this order intentionally to guide the reader from observed cellular phenotypes toward deeper mechanistic insights.

To improve clarity, we have strengthened the transitional statements between sections to better highlight this progression and ensure a smoother narrative flow.

We appreciate the reviewer's constructive comment, which helped us carefully reconsider and refine the presentation of our results.

7. Coming back to gene variants in CNO versus arterial tortuosity syndrome. Would it be possible to test a variant identified in arterial tortuosity as well to understand potential differences on inflammatory signaling and osteoclast function?

Response: We sincerely thank the reviewer for this insightful and thought-provoking suggestion.

Previously reported *SLC2A10* variants associated with arterial tortuosity syndrome (ATS) are predominantly loss-of-function mutations, including nonsense or frameshift variants that result in markedly reduced or absent GLUT10 activity. In contrast, the variant identified in our study appears to exhibit a gain-of-function effect associated with altered bone metabolism and inflammatory signaling.

Given these fundamentally different mutational mechanisms, we speculate that loss-of-function and gain-of-function variants may perturb distinct biological pathways, potentially leading to divergent clinical phenotypes. While directly comparing an ATS-associated variant with the variant identified in our study would indeed provide valuable mechanistic insight, such experiments are beyond the scope of the current work.

We agree that comparative functional analysis at the cellular level represents an important future direction and may further clarify genotype–phenotype relationships within *SLC2A10*-associated disorders.

We appreciate the reviewer's constructive suggestion, which highlights an important avenue for future investigation.

Referee #3 (Comments on Novelty/Model System for Author):

1. The main conclusions are based on a small family with only three patients. Such a small sample size makes it difficult to completely rule out the possibility that this rare variant is familial idiopathic or a chance co-segregation with the disease. The authors identified other candidate genes in two other families, but did not explore the presence of SLC2A10 mutations in these families or rule out their role, which weakens the assertion that SLC2A10 is a major pathogenic gene.

Response: We sincerely thank the reviewer for this important comment.

We acknowledge that the study is based on a single family with three affected individuals, and that the limited sample size represents an inherent limitation. Rare variant co-segregation in small pedigrees must be interpreted cautiously.

In light of the reviewer's concern and a related suggestion from another reviewer, we have removed the sequencing data from the other two families to avoid potential confounding and to maintain a focused narrative. The revised manuscript now concentrates exclusively on the well-characterized family carrying the SLC2A10 variant.

In rare disease research, identification of a candidate pathogenic variant from a small pedigree is a commonly accepted starting point, particularly when supported by comprehensive phenotypic characterization and functional validation. In our study, the SLC2A10 variant demonstrates consistent segregation within the family and is further supported by mechanistic data at the cellular level, strengthening the evidence for its pathogenic relevance in this context.

We agree that additional cohorts and independent replication will be essential to determine the broader contribution of SLC2A10 to CNO. We have revised the manuscript to avoid overstating its role as a "major pathogenic gene" and instead present it as a potential disease-associated gene requiring further validation.

We appreciate the reviewer's thoughtful critique, which has helped us clarify the scope and strength of our conclusions.

2. Although this study characterized the mutation as "gain of function," it lacks experimental data in cell or animal models demonstrating that knocking down or eliminating endogenous *SLC2A10* does not produce a phenotype similar to that of mutant overexpression to support the "gain of function" assessment. Further experiments are needed.

Response: We sincerely thank the reviewer for this important and insightful comment.

To directly address this concern, we performed additional experiments in RAW264.7 cells. Specifically, endogenous *SLC2A10* was knocked out, followed by re-expression of either wild-type or mutant *SLC2A10*. We then reassessed the inflammatory phenotype, osteoclast differentiation, and associated signaling pathways.

Our results show that *SLC2A10* knockout alone does not recapitulate the phenotype observed with mutant overexpression. In contrast, re-expression of the mutant construct restores and enhances the inflammatory and osteoclast-related alterations, whereas wild-type re-expression does not produce the same effect. These findings support the classification of the identified variant as a gain-of-function mutation rather than a simple loss-of-function effect.

The corresponding in vitro experiments are now presented in Expanded View Figure EV3, whereas the in vivo data (formerly Figure 7) have been completely replaced by newly generated experimental results in the revised version.

We appreciate the reviewer's suggestion, which has significantly strengthened the mechanistic basis of our conclusions.

3. It is noted that the manuscript contains data on human sample staining, this study needs to supplement the description of sample collection methods and relevant ethical approvals.

Response: We sincerely thank the reviewer for highlighting this important point.

We apologize for the incomplete description in the original manuscript. The revised version now includes detailed information on human sample collection and ethical approvals. Written informed consent was obtained from all subjects and family members. The study was approved by the Biomedical Ethics Committee of Peking University School and Hospital of Stomatology (Approval No. PKUSSIRB-202272025) and conducted in accordance with the Declaration of Helsinki (1964).

In addition, all animal experiments were approved by the Experimental Animal Ethics Committee of Peking University Health Science Center (Animal Protocol No. DLASBE0528).

These details have been added to the Methods section. Additional methodological information is provided in the Appendix Supplementary Methods.

We appreciate the reviewer's attention to ethical transparency.

4. The biological models used in this study were inconsistent, including human samples, mouse-derived RAW264.7 cells, and zebrafish animal models. Although this gene mutation is highly conserved, the use of multiple species models reduces the persuasiveness of the experiment. Please explain this.

Response: We thank the reviewer for raising this important methodological concern.

First, the human samples establish clinical relevance and confirm that the inflammatory and skeletal phenotypes occur in the native genetic background of affected individuals. However, patient material does not allow mechanistic manipulation.

Second, RAW264.7 macrophage-derived cells were selected because CNO is fundamentally a disorder of innate immune dysregulation and osteoclast activation. This murine system enables precise genetic manipulation (knockout and rescue experiments) and controlled interrogation of inflammatory signaling and osteoclast differentiation. Such mechanistic dissection is not feasible in primary patient samples.

Third, the zebrafish model was employed to evaluate in vivo skeletal consequences in an intact vertebrate organism. Zebrafish provide a well-established platform for studying bone development and inflammatory bone pathology, and SLC2A10 is evolutionarily conserved in its key functional domains. Importantly, the in vivo model allows assessment of organism-level phenotypes that cannot be recapitulated in isolated cell systems. In future studies, we plan to further investigate the role of SLC2A10 using mouse models to validate and extend these findings in a mammalian system.

We appreciate the reviewer's comment, which allowed us to articulate the conceptual framework underlying our model selection.

5. It is suggested that the D292E mutation alters the glucose transport activity of GLUT10. Structural simulations suggest this possibility, but direct experimental evidence, such as direct glucose uptake assays to compare the transport rate of WT and MT GLUT10 in macrophages is lacking.

Response: We sincerely thank the reviewer for this important suggestion.

We agree that structural simulations alone are insufficient to demonstrate altered transporter function. In the original study, Seahorse extracellular flux analysis revealed enhanced glycolytic activity in cells expressing the D292E variant, indicating altered metabolic flux. However, we acknowledge that glycolytic rate measurements do not directly assess glucose transport activity.

To directly address this concern, we performed additional glucose uptake assays in RAW264.7 cells following SLC2A10 knockout and re-expression of either wild-type or D292E mutant constructs. These experiments demonstrate that cells expressing the D292E variant exhibit significantly altered glucose uptake compared to wild-type controls, supporting the conclusion that the mutation affects GLUT10 transport function.

The corresponding data have been added to the EV2B.

We appreciate the reviewer's insightful comment, which has strengthened the functional validation of our conclusions.

EV3 C: Glucose uptake assay demonstrating a significantly higher glucose uptake rate in D292E macrophages compared with WT and empty vector (EV) controls. p-values: *p < 0.05, **p < 0.01, ***p < 0.001; ns: no significant difference.

6. This study has observed the simultaneous occurrence of enhanced glucose metabolism and increased inflammation/osteoclast differentiation, but have failed to demonstrate that the former is the cause of the latter. The key experiment is to determine whether the pro-inflammatory and osteoclast-promoting effects induced by mutant *SLC2A10* are reversed after the use of glycolysis inhibitors. Under the current circumstances, the "glycolysis-driven" conclusion appears to be more of a correlation than a causal relationship.

Response: We sincerely thank the reviewer for this insightful comment regarding causality. We agree that the coexistence of enhanced glycolysis and increased inflammatory/osteoclast phenotypes does not by itself establish a causal relationship. To directly address this issue, we performed additional rescue experiments.

Specifically, in SLC2A10 knockout RAW264.7 cells re-expressing the D292E mutant, we treated cells with the glycolysis inhibitor 2-deoxy-D-glucose (2-DG). Importantly, inhibition of glycolysis significantly attenuated the elevated inflammatory cytokine production and reduced osteoclast differentiation induced by the mutant construct.

These findings demonstrate that glycolytic activity is functionally required for the pro-inflammatory and osteoclast-promoting effects of the D292E variant, thereby providing causal support for a glycolysis-dependent mechanism rather than a simple correlation.

The corresponding data have been included in the Expanded View Figure (EV2C-G).

We appreciate the reviewer's critical suggestion, which has substantially strengthened the mechanistic conclusions of our study.

7. In zebrafish, it is recommended to supplement with experiments, such as knocking down endogenous *slc2a10* in zebrafish, or constructing a point mutation knock-in model using CRISPR/Cas9 to avoid the off-target effects brought by mRNA overexpression.

Response: We sincerely thank the reviewer for this important suggestion regarding the zebrafish model.

We agree that mRNA overexpression alone may introduce potential non-physiological effects. To address this concern, we performed additional experiments in zebrafish embryos using a combined loss- and gain-of-function strategy. Specifically, endogenous *slc2a10* was suppressed during early embryogenesis, followed by co-injection of either wild-type or D292E mutant mRNA. We then assessed inflammatory markers and osteoclast-related phenotypes.

Our results demonstrate that re-expression of the D292E mutant, but not the wild-type construct, restores and enhances the inflammatory and osteoclast-associated phenotypes. This rescue-based design helps distinguish mutation-specific effects from generalized overexpression artifacts.

The corresponding data have been updated in Figure 7.

Figure 7. Zebrafish model injected with SLC2A10 mRNA and mutated mRNA. (A) The mRNA expression of inflammatory and osteoclast markers at 4 days post-fertilization in four groups. (B) Quantification of ROS fluorescence intensity in Control, KO, WT and D292E groups. (C) Neutral red staining marks macrophages in two groups of mandibles in 12 dpf zebrafish. (The white dashed line delineates the jawbone region. Macrophages are shown in red and are indicated by yellow arrows.) (D) Detection of mineralization effect by alizarin red staining in 21 dpf zebrafish. CTL: healthy control zebrafish; KO: SLC2A10 knockout zebrafish; WT: zebrafish model injected with SLC2A10 mRNA; D292E: zebrafish model injected with mutated mRNA. N = 10 per group. One-way ANOVA was used to evaluate the significance between multiple groups and Dunnett test was used to calculate the P-value for post-hoc comparisons. All data are shown as mean \pm SD; *P < 0.05, **P < 0.01, ***P < 0.001, ****P < 0.0001.

We acknowledge that generation of a CRISPR/Cas9 knock-in model would provide an additional level of genetic precision. However, establishment and validation of a stable knock-in line require substantial time and are beyond the scope of the current study. We consider this an important direction for future investigation.

8. This study only described the phenotypes of macrophage polarization, osteoclast differentiation, and the effects on osteoblasts, without explaining the mechanisms involved. For example, the mechanisms of inflammatory osteoclast differentiation and the intermediate mediators in osteoclast/osteoblast co-culture were not further explored, which reduced the depth of the study.

Response: We sincerely thank the reviewer for this constructive comment regarding the mechanistic depth of our study.

To address this concern, we performed additional mechanistic analyses focusing on osteoclast differentiation. Our new data demonstrate that cells expressing the D292E variant exhibit enhanced activation of the ERK–AP1 signaling pathway during osteoclastogenesis, as evidenced by increased ERK phosphorylation and downstream transcriptional activation. In contrast, wild-type cells did not show elevated ERK phosphorylation under the same conditions.

Importantly, treatment with the glycolysis inhibitor 2-deoxy-D-glucose (2-DG) significantly reduced ERK activation and reversed the upregulation of osteoclast-related factors. These findings indicate that the mutant-induced enhancement of osteoclast differentiation is mediated, at least in part, through a glycolysis-dependent activation of the ERK–AP1 pathway.

The corresponding data have been included in the EV2F-G, and the mechanistic description has been expanded in the revised Results and Discussion sections.

We appreciate the reviewer's valuable suggestion, which has strengthened the mechanistic framework of our study.

We appreciate the reviewer's constructive suggestion, which has strengthened the in vivo validation of our findings.

EV3 (F) RANKL stimulation for 2 days significantly activated ERK signaling in D292E cells, and this activation was reduced by 2-DG treatment. (G) RANKL stimulation for 2 days, the AP-1 component c-JUN was upregulated in D292E cells, and this effect was partially reversed by 2-DG.

9. The grammar and accuracy of the manuscript need to be thoroughly corrected, such as the standardization of gene naming.

Response: We sincerely thank the reviewer for this important comment.

We agree that grammatical accuracy and standardized gene nomenclature are essential for clarity and scientific rigor. The revised manuscript has undergone thorough language editing, including correction of grammatical errors and standardization of gene and protein nomenclature in accordance with accepted conventions.

We are grateful for the reviewer's attention to detail, which has significantly improved the precision and presentation of our work.

May 18, 2026

RE: Life Science Alliance Manuscript #LSA-2026-03772-T

Prof. Jin Gang An
Peking University School and Hospital of Stomatology
Department of Oral and Maxillofacial Surgery
China

Dear Dr. An,

Thank you for submitting your revised manuscript entitled "A Novel SLC2A10 Gain-of-Function Variant Links Glycolytic Macrophage Polarization to Chronic Nonbacterial Osteomyelitis". In accordance with the offer conveyed by our partners at another journal, and as confirmed in our subsequent correspondence, we are glad to consider this work for publication in Life Science Alliance. As previously noted, publication is dependent on the following changes to the text to more closely align the claims with the evidence provided:

- Temper the claims based on representative images of neutral red staining and Alizarin Red staining in the Results and Discussion and associated figure legends, per Reviewer 5.
- Include in the introduction and discussion the prior observations on ER localization and ascorbate transport activity of GLUT10, per Reviewer 5.
- In view of the comments by Reviewer 4, adjust the first paragraph in the discussion to acknowledge that the "new mechanistic framework" provided in this work still requires validation.

Because this work includes photographs of minors, please include a statement in the Study Approval section that the parent guardians have approved the publication of these photos via written consent. Please then complete the attached LSA Ethics Approval form (by copying the completed Study Approval section) and upload the signed form to our system as a manuscript file.

Finally, please also tend to the points noted below, which are required to align with journal formatting policies. We will be happy to publish your paper in Life Science Alliance pending completion of these text and formatting changes.

MANUSCRIPT ORGANIZATION AND FORMATTING:

To avoid unnecessary delays in the acceptance and publication of your paper, please read the following information carefully. Full guidelines are available on our Instructions for Authors page, <https://www.life-science-alliance.org/authors>

- LSA allows supplementary figures, but not EV Figures; please update your callouts for the Supplementary Figures in the manuscript (Fig EV1A = Fig S1A) and everywhere else in the text.
- Please be sure that the authorship listing and order are correct and match between the system and the manuscript file.
- Please add a title page to your manuscript file. The full name (middle names as initials) of each author should be given. Multiple first-authorships are acceptable and should be indicated. Numbers in superscript should be used to indicate the department, institution, city, and country for each author. Any changes of address may also be given in numbered footnotes.
- Please mark the corresponding authors on the manuscript file as well.
- Please add ORCID IDs for all corresponding authors - you should have received instructions on how to do so.
- The Abstract should not exceed 175 words, and has to match between the system and the manuscript file.
- Please add a Summary Blurb/Alternate Abstract in our system.
- Please add Keywords and a Category for your manuscript in our system.
- Please add the X and Bluesky handles of your host institute/organization, as well as your own, and/or one of the authors, in our system.
- Please also add the Author Contributions to our system.
- Please use the [10 author names et al.] format in your references (i.e., limit the author names to the first 10).
- Please separate the Figure legends and Supplemental Figure legends into separate sections.
- Please move the methods details in the Appendix file to the main Materials and Methods section.
- Please provide a publicly accessible accession number for the RNA seq data in the Data Availability section.
- There is a call-out for figure S5, and this figure doesn't exist - please correct.
- Please add callouts for Figures S2A-D and S3A-G to your main manuscript text.

spreadsheets for the main figures of the manuscript. If you would like to add source data, we would welcome one PDF/Excel-file per figure for this information. These files will be linked as supplementary "Source Data" files.

We welcome submissions of potential cover images for the issue of LSA in which your work would appear. If you have high quality images associated with this work, please feel free to email these, with a caption, to the journal office.

LSA encourages authors to provide a 30-60 second video where the study is briefly explained. These videos will be appear embedded with the manuscript online at Life Science Alliance, and on social media to promote the published paper and authors (for examples, see <https://docs.google.com/document/d/1-UWCfbE4pGcDdcgzcmiuJl2XMBJnxKYeqRvLLrLS08s/edit?usp=sharing>). Corresponding or first-authors are welcome to submit the video. Please submit only one video per manuscript. The video can be emailed to contact@life-science-alliance.org

FINAL FILES:

The following items are required for acceptance.

The license to publish form must be signed before your manuscript can be sent to production. A link to the license to publish form will be available to the corresponding author only. Please take a moment to check your funder requirements.

Thank you for your attention to these final processing requirements. Please revise and format the manuscript and upload materials as soon as you are able.

Thank you for this interesting contribution to the literature. We look forward to publishing your paper in Life Science Alliance.

Sincerely,

May 27, 2026

RE: Life Science Alliance Manuscript #LSA-2026-03772-TR

Prof. Jingang An
Peking University Stomatological Hospital
Department of Oral and Maxillofacial Surgery
22# Zhongguancun South Avenue
Beijing
China

Dear Dr. An,

Thank you for submitting your Research Article entitled "A SLC2A10 Gain-of-Function Variant Links Macrophage Glycolysis to Chronic Nonbacterial Osteomyelitis". It is a pleasure to let you know that your manuscript is now accepted for publication in Life Science Alliance. Congratulations on this interesting work. We will endeavor to complete proofing and publication online as fast as possible.

Your article will publish open access upon publication under a CC-BY license.

DISTRIBUTION OF MATERIALS:

Again, congratulations on a very nice paper. I hope you found the review process to be constructive and are pleased with how the manuscript was handled editorially. We look forward to future exciting submissions from your lab.

Sincerely,
